# Measurement Report: Combined use of MAX-DOAS and AERONET ground-based measurements in Montevideo, Uruguay for the detection of distant biomass burning

Matías Osorio[1], Alejandro Agesta[1], Tim Bösch[2,3], Nicolás Casaballe[1], Andreas Richter[2], Leonardo M. A. Alvarado[2,3], and Erna Frins[1]

[1]Instituto de Física, Facultad de Ingeniería, Universidad de la República, Montevideo, Uruguay
[2]Institute of Environmental Physics, University of Bremen, Bremen, Germany
[3]Alfred Wegener Institute, Helmholtz Centre for Polar and Marine Research, Bremerhaven, Germany

**Correspondence:** Matías Osorio (mosorio@fing.edu.uy), Erna Frins (efrins@fing.edu.uy)

**Abstract.** Biomass burning releases large amounts of aerosols and chemical species into the atmosphere, representing a major source of air pollutants. Emissions and by-products can be transported over long distances, presenting challenges in quantification. This is mainly done using satellites, which offer global coverage and data acquisition for places that are difficult to access. In this study, ground-based observations are used to asses the abundance of trace gases and aerosols. On November 24, 2020, a significant increase in formaldehyde was observed with a Multi-AXis Differential Optical Absorption Spectroscopy (MAX-DOAS) instrument located in Montevideo (Uruguay) and its vertical column densities reached values of $2.4 \times 10^{16}$ molec. $\mathrm{cm}^{-2}$, more than twice the values observed during the previous days. This was accompanied by an increase in the aerosol levels measured by an AErosol RObotic NETwork (AERONET) photometer located at the same site. The Aerosol Optical Depth (AOD) at 440 nm reached values close to 1, one order of magnitude larger than typical values in Montevideo. Our findings indicate that the increase was associated with the passage of a plume originating from distant biomass burning. This conclusion is supported by TROPOspheric Monitoring Instruming (TROPOMI) satellite observations as well as HYbrid Single-Particle Lagrangian Integrated Trajectory (HYSPLIT) simulations. The profiles of the gases and aerosols retrieved from the MAX-DOAS observations are consistent with the HYSPLIT analysis, showing the passage of a plume over Montevideo on November 24 located at a height of $\sim$1.5 km. This corroborates that biomass burning events occurring about 800 km north of Montevideo can affect the local atmosphere through long-distance emissions transport. This study underscores the potential of ground-based atmospheric monitoring as a tool for detection of such events. Furthermore, it demonstrates greater sensitivity compared to satellite when it comes to detection of relatively small amounts of carbonyls like glyoxal and formaldehyde.

## 1 Introduction

Biomass Burning (BB) events significantly impact air quality, both regionally and globally due to the direct emissions of various substances into the atmosphere and the production of species during the subsequent long-distance transport. This issue and its impact on cities has been addressed in several studies using satellite observations, which offer the advantage of broad

spatial coverage compared with ground-based measurements (Wittrock et al., 2006; Zarzana et al., 2017; Alvarado et al., 2020; Schutgens et al., 2021; Lerot et al., 2023).

In addition to producing aromatic hydrocarbons, nitrogen oxides ($NO_x$), carbonyls, and organic carbon, fires are the main source of aerosols containing particulate matter (PM), generating peaks of very high PM10 concentrations during the dry seasons (de Oliveira Alves et al., 2015). Furthermore, the pollutants produced during these fires can be transported over long distances, affecting areas far away from their source (Hsu et al., 1996; Freitas et al., 2005; Ravindra et al., 2008).

Aerosols have adverse effects on human health in exposed populations (Chen et al., 2017; Contini et al., 2021), leading to respiratory diseases. Additionally, aerosols originating from BB contain soot, which absorbs solar radiation, thus altering the radiative budget of the atmosphere, cloud formation processes, and the albedo of snow and ice covered areas (Seinfeld and Pandis, 2006; Bond et al., 2013; Bellouin et al., 2020).

In South America, thousands of square kilometers of the Amazon are affected by burning to carry out agricultural, livestock, mining, and infrastructure projects each year. Uncontrolled fires of other types of vegetation also occur, for example, in Argentina, in the Paraná delta, and in the border region between Argentina, Brazil and Paraguay (Gassmann and Ulke, 2008).

Uruguay is located south of Brazil and east of Argentina with its capital, Montevideo, positioned about 450 km from the Paraná delta (Argentina) and approximately 800 km from the tri-border region between Argentina, Paraguay, and Brazil. Large fires occur frequently in both regions, affecting air quality in Montevideo. These fires can be identified in satellite images due to the presence of large amounts of smoke. Air quality in Uruguay is generally good (Ministerio de Ambiente, República Oriental del Uruguay, 2024; Intendencia de Montevideo, 2024); however, these fires induce significant short-term changes in the lower atmosphere, altering its composition.

Since 2020, a CIMEL sunphotometer of the AErosol RObotic NETwork (AERONET, led by NASA – Goddard Space Flight Center) and a Multi-AXis Differential Optical Absorption Spectroscopy (MAX-DOAS) system of the Applied Optics Group of the Institute of Physics at the Engineering Faculty (UDELAR) have been operational to perform ground-based continuous monitoring of atmospheric constituents over Montevideo.

DOAS has become a widely used technique to study the atmosphere remotely, due to its versatility and adaptability, allowing a broad range of configurations (Platt and Stutz, 2008; Rivera et al., 2010; Ibrahim et al., 2010; Shaiganfar et al., 2017). Applying DOAS, the main sources of $SO_2$ and $NO_2$ emissions from industrial activities in Montevideo have been studied, such as the "La Teja" oil refinery (Frins et al., 2012; Osorio et al., 2017) and also the "Central Batlle y Ordoñez" power plant (Frins et al., 2014). Despite these local sources of emissions, the air quality in Montevideo remains at good levels, due to its location at the coast of the La Plata River and the prevailing winds. Although the combination of ground-based instruments such as DOAS with AERONET is widely used in the northern hemisphere for detecting and analyzing emissions from biomass burning events, its application is quite rare in the southern hemisphere, especially in South America.

The objective of this study is to explore the use of ground-based observations for the detection of aerosols and trace gases produced by distant BB emissions transported over long distances to Montevideo. In particular, we investigated an event that occurred in November 2020 on the border between Paraguay and Argentina, situated approximately 800 km north of

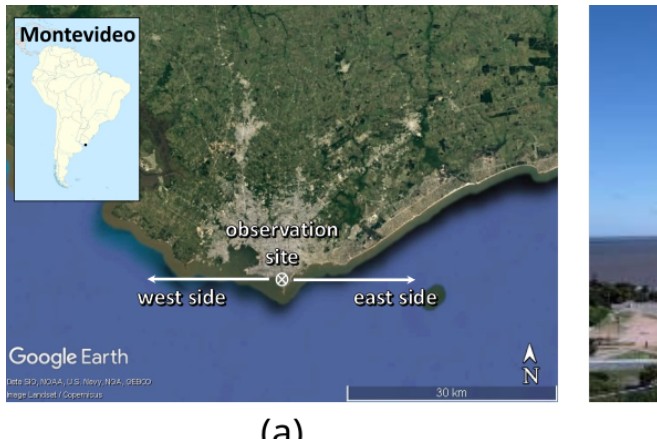
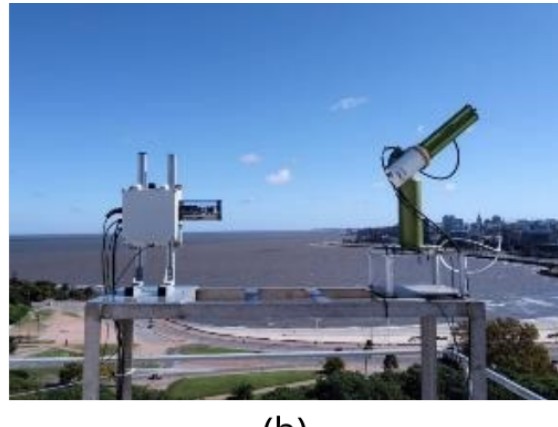

(a)            (b)

**Figure 1.** (a) Map of Montevideo and the geometry used for the MAX-DOAS measurements. The observation site is located at the Faculty of Engineering. The spectrometer scans a vertical plane from east to west at several elevations. (b) Picture of the MAX-DOAS instrument (left) and the CIMEL sun photometer (right) on the roof of the building.

Montevideo. This relatively strong event resulted in the detection of a plume passing over Montevideo, by means of multiple instruments covering ground-based and satellite observations.

This article is organized as follows: in Section 2 we describe the methods used for monitoring the event and the additional tools employed to support our conclusions; in Section we 3 present the DOAS observations and additional results used for detecting the event in November 2020; in Section 4 we present a discussion of our findings; in Section 5 we provide some general remarks and the conclusions from this work.

## 2 Instruments and methods

### 2.1 Site description

In this study, a MAX-DOAS system and a sun photometer from the AERONET program were used for remote sensing of the atmosphere. Both instruments are installed on the roof of the Faculty of Engineering building located in Montevideo, Uruguay (34.9175° S, 56.1669° W). The MAX-DOAS has been providing data since November 2020, and the sun photometer since January 2020. Figure 1(a) indicates the location of the instruments and the MAX-DOAS observation directions on the map. Figure 1(b) shows a picture of both instruments on the roof of the building. The main characteristics of each instrument are described below.

### 2.2 MAX-DOAS instrument

A MAX-DOAS instrument manufactured by AirYX GmbH operates continuously and acquires diffuse solar radiation spectra at several elevation angles in a vertical plane from east to west. The spectra are acquired by a thermo-controlled Avantes

spectrometer that operates in the spectral range of 301-463 nm and has a spectral resolution of approximately 0.6 nm. The total field of view of the system is 0.3 degrees in the vertical direction and 1 degree in the horizontal. The MAX-DOAS instrument includes a mercury lamp for wavelength calibration. Dark current and offset measurements are acquired at night to correct the spectra. The temperature of the spectrometer was set to 20°C.

For this study we considered elevation angles of 1, 2, 3, 5, 10, 20, 40, and 90 degrees for both western and eastern viewing directions of observation (see Figure 1(a)). The acquisition time of each spectrum was one minute, and the duration of each vertical plane scanning sequence was approximately 20 minutes.

## 2.3 AERONET sun photometer

The AERONET node at Montevideo uses a CIMEL CE3128-T multispectral photometer (Holben et al., 1998) installed adjacent to the MAX-DOAS instrument. This photometer measures direct and diffuse solar radiation at different bands ranging from 340 to 1640 nm to retrieve aerosol data from the observations. AERONET provides several data products derived from these measurements, e.g. aerosol optical depth (AOD), single scattering albedo (SSA) and phase function (Holben et al., 2001; Dubovik et al., 2000; Dubovik and King, 2000). In this study we used AERONET V3 level 2.0 data, ensuring good quality of data by applying radiometric and instrumental corrections, as well as cloud-cover removal algorithms (Sinyuk et al., 2020).

## 2.4 Satellite observations

The TROPOspheric Monitoring Instrument (TROPOMI), carried by the Copernicus Sentinel-5 Precursor satellite, was successfully launched on October 13, 2017. It operates within the ultraviolet (UV), visible (VIS), near-infrared (NIR) and short-wave infrared (SWIR) spectral range, covering wavelengths from 270 to 500 nm in the UV-VIS, from 675 to 775 nm in the NIR, and a SWIR band from 2305 to 2385 nm.

TROPOMI provides near-global daily coverage, achieving a spatial resolution of 3.5 km $\times$ 5.5 km (7 km $\times$ 7 km in the SWIR) since August 2018. It crosses the equator at 13:30 local time (ascending node). Similar to the Ozone Monitoring Instrument (OMI), TROPOMI uses a nadir-viewing imaging spectrograph with a two-dimensional charged-couple device (CCD). One dimension captures spectral information, while the other dimension covers the spatial information. The TROPOMI instrument has been delivering data since November 2017 (Veefkind et al., 2012). Its spectral bands enable the observation of relevant atmospheric species such as $CHOCHO$, $HCHO$, $NO_2$, and $CO$.

We employed TROPOMI observations of $NO_2$ and $CO$ in our analysis to detect long distance transport events. We also used UV Aerosol Index (UVAI) data as a reference for the presence of aerosols in the plume, which likely coincides with the location of the trace gases (see for instance Torres et al. (2020); Alvarado et al. (2020)). These products have been operational since March 2018, and they include the reprocessed (RPRO) and offline-mode (OFFL) data versions OFFL V01.03.02, RPRO V02.04.00, OFFL V01.03.02, respectively. For the retrieval of formaldehyde (HCHO) and glyoxal (CHOCHO), volatile organic compounds (VOC), the products considered in this study are based on (Alvarado et al., 2020). Glyoxal has been retrieved in the wavelength range from 433 to 465 nm, which is slightly larger than windows used in previous investigations (Vrekoussis et al., 2010; Alvarado et al., 2014). HCHO is retrieved in the fitting window from 323.5 nm to 361 nm, which results in a re-

duction of the noise of slant column density. The main interfering absorbers are included in the respective wavelength ranges as described in (Alvarado et al., 2020). To compute Vertical Column Densities (VCD), airmass factors based on trace gas profiles simulated with the Tracer Model 5 - Massively Parallel version (TM5-MP) global chemistry transport model are applied to the retrieved slant column density (SCD) data (Myriokefalitakis et al., 2020). For an individual CHOCHO measurement, the detection limit is of the order of $5 \times 10^{14}$ molec. cm$^{-2}$. For HCHO, the detection limit is of the order of $4.5 \times 10^{15}$ molec. cm$^{-2}$.

## 2.5 HYSPLIT model

The Hybrid Single-Particle Lagrangian Integrated Trajectory (HYSPLIT) model, developed by the National Oceanic and Atmospheric Administration (NOAA), is used to track emissions of gases and aerosols into the atmosphere. It is used to model trajectories of air mass parcels, as well as to simulate transport, dispersion, chemical processes, and deposition (Stein et al., 2015). HYSPLIT uses a stratified atmosphere and relies on three-dimensional weather data, such as wind speed and direction, temperature, and humidity. These weather data can be obtained from ground-based weather stations, weather balloons, satellites, or numerical weather models.

In this study, we used HYSPLIT to simulate the trajectories of air parcel originating from distant biomass burning sites. This approach allowed for the evaluation of the feasibility of long-distance plume transport (see Section 3.2).

## 2.6 BOREAS

The Bremen Optimal estimation REtrieval for Aerosols and trace gaseS (BOREAS, Bösch et al., 2018) is an inversion algorithm designed for retrieving vertical profiles of trace gases and aerosol from MAX-DOAS measurements of differential Slant Column Densities (dSCD). This algorithm was developed at the Institute for Environmental Physics in Bremen, Germany (Bösch et al., 2018) and has been validated in various studies (e.g. Frieß et al., 2019; Tirpitz et al., 2021). Each input vector includes dSCD measurements taken at different lines of sight for either the trace gas of interest or the oxygen collision complex $O_4$. The latter has a well-known vertical profile in the atmosphere, and its slant columns can therefore be used to retrieve the vertical distribution of aerosols.

In this study, aerosols have been retrieved from the $O_4$ absorption band at 360 nm. Using the mean Ångström Exponent 440 - 870 nm from AERONET Inversion data (Version 3, Almucantar, Level 1.5), the retrieval results have been extrapolated to the matching spectral windows of HCHO and CHOCHO, as the AERONET station only retrieves data at 440 nm or higher wavelengths. AERONET Inversion data was also used for the aerosol parametrization within BOREAS. The AERONET 440 nm single scattering albedo (SSA) and phase function obtained at the time closest to the corresponding slant column measurement scan have been used within the profile inversion (440 nm products are the closest available to 360 nm wavelength products).

Atmospheric profiles over Montevideo were also simulated using the Community Atmosphere Model – Chemistry model (Buchholz et al., 2019; Emmons et al., 2020) and served as a priori meteorological data sets within BOREAS.

Each trace gas profile was retrieved by applying a pre-scaled exponential a priori profile (Bösch et al., 2018) with an exponential scale height of 1km for all species. In general, the sensitivity of MAX-DOAS profiling algorithms is highest

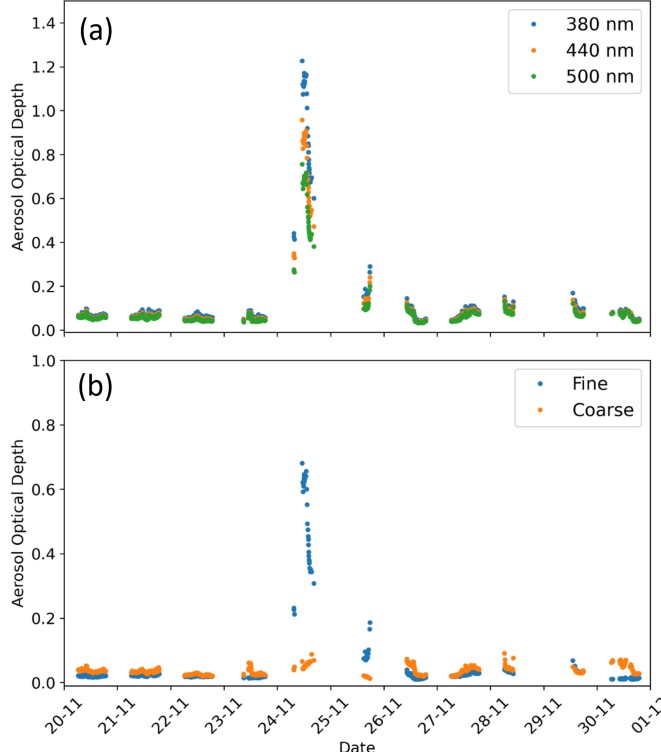

**Figure 2.** AOD detected at the AERONET station in Montevideo for the period 18 November 2020 to 30 November 2020. Panel (a) shows the AOD for the wavelength bands of 380, 440 and 500 nm. Panel (b) shows the AOD at 500 nm for fine and coarse particles separately.

for the lowest retrieval layers and decreases strongly for altitudes above 2 - 3 km. However, if an elevated trace gas layer
is dominant – with no significant trace gas concentration present below the elevated layer – the retrieval is possible, but the
vertical extent of the layer might be retrieved with slightly decreased accuracy (Bösch, 2019; Tirpitz et al., 2021).

## 3   Detection of a distant biomass burning event

### 3.1   Signals of the biomass burning event

AERONET data provided early indications of the presence of a biomass burning plume over Montevideo in November of
2020. Aerosol optical depth (AOD) values usually range around $0.09 \pm 0.03$ in the 440 nm band when measured at this station.
However, on November 24, AOD values reached up to 0.96 from the photometer measurements, indicating the presence of
large amounts of aerosols in the atmosphere.

   Figure 2(a) shows AOD values for a period around November 24 at wavelengths 380, 440 and 500 nm and Figure 2(b) shows
the contribution of the fine and coarse modes to the AOD at 440 nm (O'Neill et al., 2003). The prevalence of fine particles

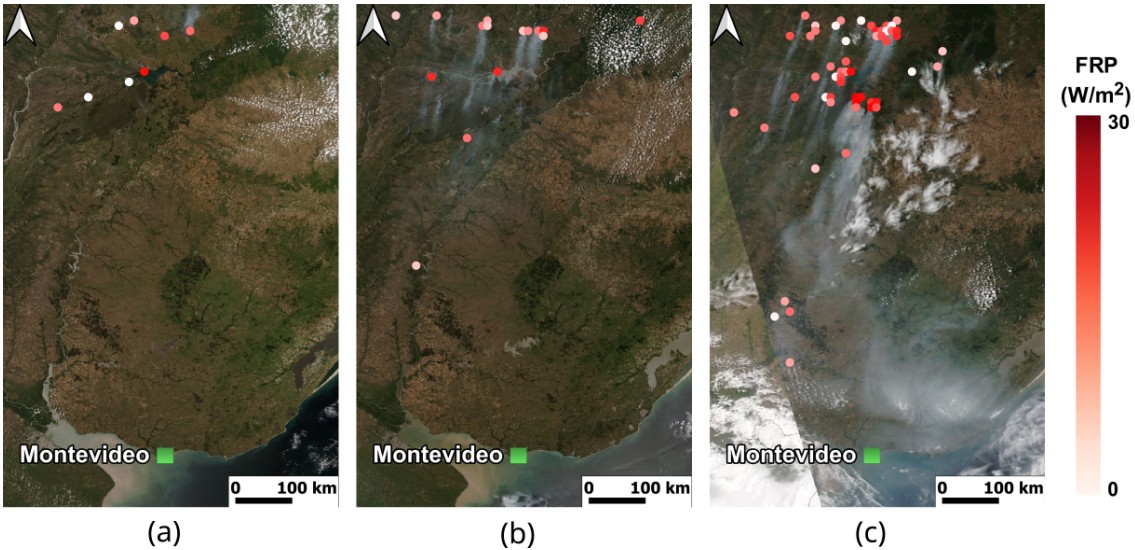

**Figure 3.** RGB satellite images from the VIIRS instrument (NOAA-20) for days (a) November 22, (b) November 23, and (c) November 24, 2020. The corresponding fire radiative power (FRP) data product was overlaid to visualize the location of the principal burning sources.

(those with radius smaller than 1 $\mu$m) is interpreted as an indicator of a biomass burning plume passing above Montevideo (Agesta, 2023). Notably, the AOD due to fine particles increased on November 24 with respect to the average of the period ($0.46 \pm 0.06$ versus $0.03 \pm 0.01$). This increase is a characteristic of biomass burning smoke (Shi et al., 2019).

Analyzing RGB satellite images from the VIIRS instrument (NOAA-20) for dates around November 24, the presence of an extensive plume over a large region in the central-eastern Uruguayan territory was detected. The origin of this plume can be attributed to several distant emission sources primarily related to biomass burning, as shown in Figure 3. Some fire spots appear near the border between Paraguay and Argentina, approximately 800 km north of Montevideo, around November 22. The number of foci observed increased towards November 23, and a distinguishable plume was transported southwards over Uruguay on November 24. As a proxy for identifying fire location and fire intensity, the Fire Radiative Power (FRP) data product from Copernicus Atmosphere Service Information is superimposed in Figure 3, indicating the locations of the fires (Kaiser et al., 2012).

### 3.2 Plume transport simulations with the HYSPLIT model

To investigate possible transport pathways, several simulations were conducted using the HYSPLIT trajectory model around the event date to track the transport of air mass parcels from the fires and verify if the emitted plume was expected to appear over Montevideo. We estimated the location and height of the main fire spots appearing around November 23 at 17:00 UTC using satellite data (see Figure 3).

The trajectory simulations start on November 23 at 17:00 UTC from a total of nine points, using wind field data from the Global Forecast System at 0.25 degrees horizontal resolution to drive the HYSPLIT model, which also provides the boundary

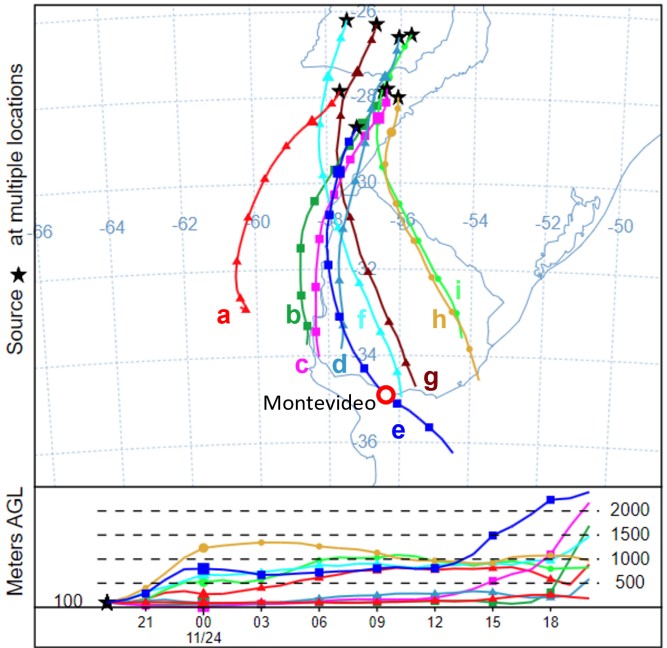

**Figure 4.** HYSPLIT simulated trajectories for fire sources located 800 km north of Montevideo detected on November 23. The line markers indicate 3 h intervals. Several trajectories are passing near Montevideo around the time of the measurements (e, f, g).

layer height. Data from the Weather Research and Forecasting model was selected as input for the simulation. This also defines the height interpolation, with a maximum height set to 15 kilometers. Figure 4 shows an example of these simulations, for which the height of each source was set at 100 m above ground level. Additional initial conditions were investigated with starting heights set to 500 and 1000 m altitude to ensure that the air parcels appear well within the boundary layer, allowing them to be transported hundreds of kilometers. The results are similar and, in all cases, air parcels from the areas surrounding the fires reached Montevideo at the time corresponding to the detection through our observations.

These trajectories indicate that a portion of the emitted plume is transported from north to south and then towards east of Montevideo on November 24 (trajectories e, f and g in Figure 4), in agreement with the rest of the observations for the same date. Other parts of the plume were transported towards the west of Uruguay, at distances exceeding 100 km from Montevideo (trajectories a-d, h and i in Figure 4).

The modeled plume remained below an altitude of 2500 m with respect to sea level (bottom panel in Figure 4). Here, we assume that aerosols and the principal derived products of the biomass burning are transported at the same altitude.

## 3.3 DOAS retrieval

The principles of the DOAS analysis are described for example in (Hönninger et al., 2004) and in (Platt and Stutz, 2008). We performed the DOAS analysis of the spectra recorded by the MAX-DOAS instrument, with primary focus on retrieving the dSCD of trace gases commonly associated with wildfire events, such as formaldehyde (HCHO), glyoxal (CHOCHO) and nitrogen dioxide ($NO_2$). Additionally, the analysis addressed the dSCD of the oxygen dimer $O_4$, which is used to quantify the presence of aerosols in the atmosphere (Wagner et al., 2004; Frieß et al., 2006). The evaluations were performed using the QDOAS software (Danckaert et al., 2017). Table 1 summarizes the settings used for the retrievals of each trace gas of interest in three fitting intervals: 333 to 359 nm for HCHO, 352 to 387 nm for $O_4$ and 432.5 to 459 nm for CHOCHO and $NO_2$. The high-resolution absorption cross sections required for the trace gas retrieval were taken from the MPI-Mainz UV/VIS Spectral Atlas (Keller-Rudek et al., 2013). The reference spectra used in the retrievals were the zenith measurements closest in time to each scanning sequence. This choice is intended to enhance the sensitivity towards tropospheric absorptions. In the DOAS retrieval, a synthetic Ring spectrum calculated with QDOAS was also included (Grainger and Ring, 1962; Chance and Spurr, 1997). A polynomial of degree 5 was used to fit broadband structures, along with allowance for a second order shift and stretch of the wavelengths of the spectra.

Figure 5 shows an example of the DOAS analysis for each fitting interval, performed on a spectrum acquired on November 24, 2020 at 13:04 local time (LT). The first row shows the fitting of the target gases, HCHO, $O_4$ and CHOCHO. Other trace gases exhibiting absorption in the same spectral interval, as well as the synthetic Ring spectrum, are shown in the second and third rows. The residual of the analysis is shown in the last row.

In this example, the DOAS fits reveal a strong formaldehyde signal, in contrast to what usually appears in Montevideo. In addition, glyoxal was also observed, which had never been detected previously in the same location. A strong signal was also observed for $NO_2$. However, this cannot necessarily be attributed to the influence of the event over Montevideo. A detailed discussion of this point is presented in Section 4.

Cloudy days were screened from our spectral data set using an algorithm based on the diurnal cycle of the color index given by the ratio between intensities at 370 nm and 440 nm (Gielen et al., 2014; Wagner et al., 2014). This ratio is strongly affected by cloudy conditions, and its variations provide a way to identify clear days. November 25 and 28 were tagged as completely cloudy and removed from the analysis as a result of this first order data screening. We used data from weather stations to manually confirm this result.

## 4 Results and discussion

### 4.1 MAX-DOAS

The results of the DOAS retrievals for the period around the detection of the event are shown in Figure 6 for the trace gases HCHO, $O_4$, CHOCHO and $NO_2$ (first to fourth rows, respectively). The colored circles correspond to the dSCD values for elevation angles of 5 (red), 10 (green) and 20 (blue) degrees in the east and west observation directions. The $O_4$ dSCDs

**Table 1.** Retrieval settings and cross sections used on each DOAS analysis.

| Target trace gas | Fitting interval | Fitted absorber | Reference |
|---|---|---|---|
| HCHO (294K) | 333.0-359.0 nm | HCHO | Meller and Moortgat (2000) |
| | | $NO_2$ (294K) | Vandaele et al. (1998) |
| | | $O_3$ (223K, 243K) | Serdyuchenko et al. (2014) |
| | | $O_4$ | Thalman and Volkamer (2013) |
| | | BrO | Fleischmann et al. (2004) |
| $O_4$ (294K) | 352.0-387.0 nm | $NO_2$ (294K) | Vandaele et al. (1998) |
| | | $O_3$ (223K) | Serdyuchenko et al. (2014) |
| | | $O_4$ | Thalman and Volkamer (2013) |
| | | BrO | Fleischmann et al. (2004) |
| CHOCHO and $NO_2$ | 432.5-459.0 nm | $NO_2$ (220K, 294K) | Vandaele et al. (1998) |
| | | CHOCHO | Rothman et al. (2013) |
| | | $H_2O$ | Rothman et al. (2013) |
| | | $O_4$ | Thalman and Volkamer (2013) |

obtained in the period from November 20 to 28 have a behavior typical of days with clear skies and low aerosol load, except for November 24. On this day, $O_4$ results are unusually low, showing approximately the same values at all elevation angles. This can be interpreted as the result of multiple scattering in the atmosphere, which alters light path lengths and indicates a
215 high aerosol load (Wagner et al., 2004; Sinreich et al., 2013).

The main interest of our DOAS analysis is to assess simple carbonyls such as formaldehyde and glyoxal that will be used as plume tracers from biomass burning events (Hoque et al., 2018; Kluge et al., 2020). Formaldehyde dSCDs showed a clear increase on November 24 compared to earlier dates. Following November 24 and 25 (the latter was a cloudy day), HCHO levels returned to the values observed before the arrival of air masses from biomass burning. A difference is also observed between
220 the dSCDs obtained from the spectra acquired in the east and west directions. This may arise because the measurements obtained in the westerly direction may be influenced by industrial and port activities around Montevideo Bay, while eastbound observations were only affected by city activity, primarily traffic.

A slight increase in retrieved glyoxal dSCDs is observed on November 24, although it is not as pronounced as for formaldehyde. The difference of this signal compared to the background is more noticeable for the spectra obtained looking in the
225 easterly viewing direction. $NO_2$ dSCDs shows a more regular behavior throughout the study period in the westerly viewing direction, but some unusually large values at 20° elevation on November 24. When analyzing the spectra obtained from observations in the easterly viewing direction, the effect is larger and again the 20° measurements are unusually high. The DOAS retrieval uncertainties range from 3% to 11% for HCHO and from 6% to 17% for Glyoxal on November 24. On other days, these uncertainties vary between 15% and 50% respectively, as tropospheric columns are much smaller.

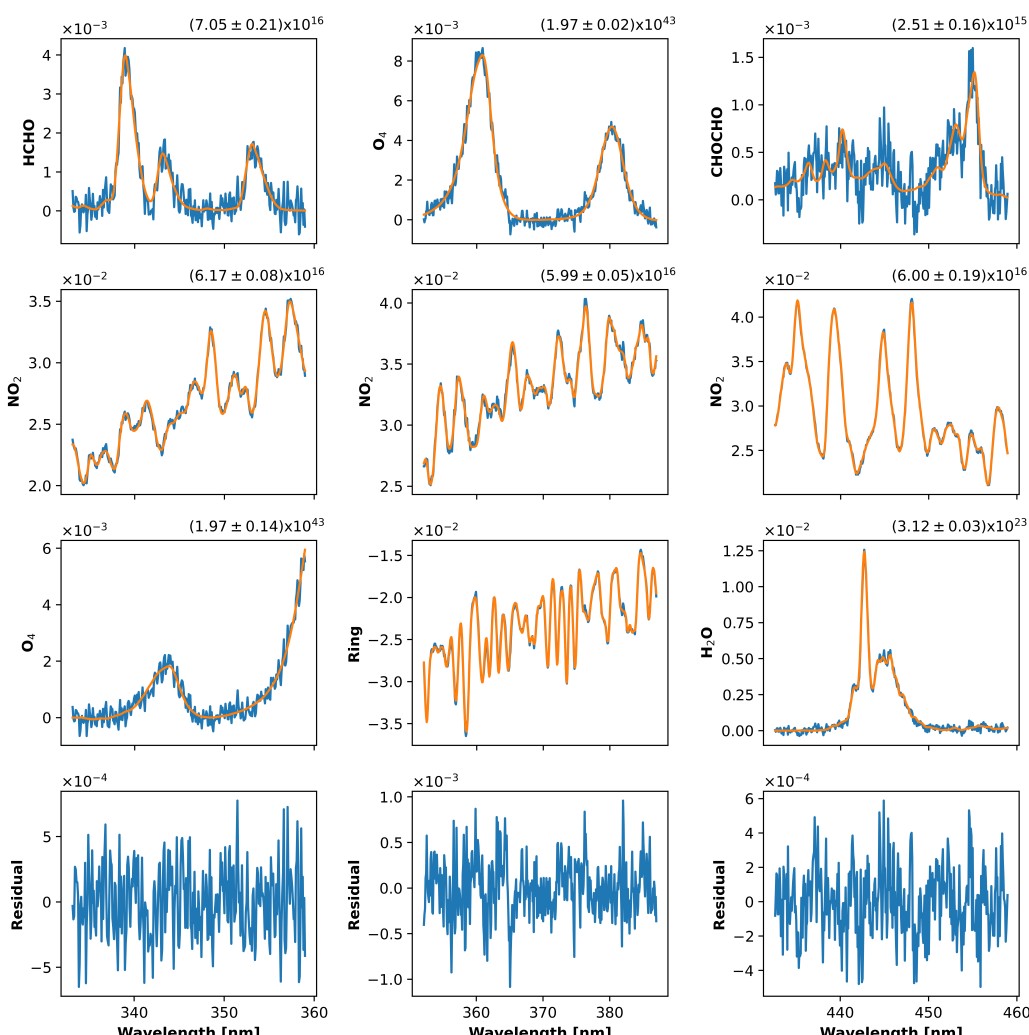

**Figure 5.** Example of DOAS fits for a spectrum measured during the event on November 24 at 13:04 LT, recorded at 5 degree elevation angle pointing towards west. Blue lines are the optical densities associated with each trace gas and the orange lines are the fitted differential absorption cross sections. The retrieved dSCD and their fitting errors are displayed in $\mathrm{molec.\,cm^{-2}}$ for HCHO, CHOCHO, $NO_2$ and $H_2O$, and in $\mathrm{molec.^2\,cm^{-5}}$ for $O_4$.

The detection limit for HCHO and glyoxal columns was estimated to be $4.0 \times 10^{15}$ molec. $\mathrm{cm^{-2}}$ and $5.6 \times 10^{14}$ molec. $\mathrm{cm^{-2}}$, respectively (Platt and Stutz, 2008).

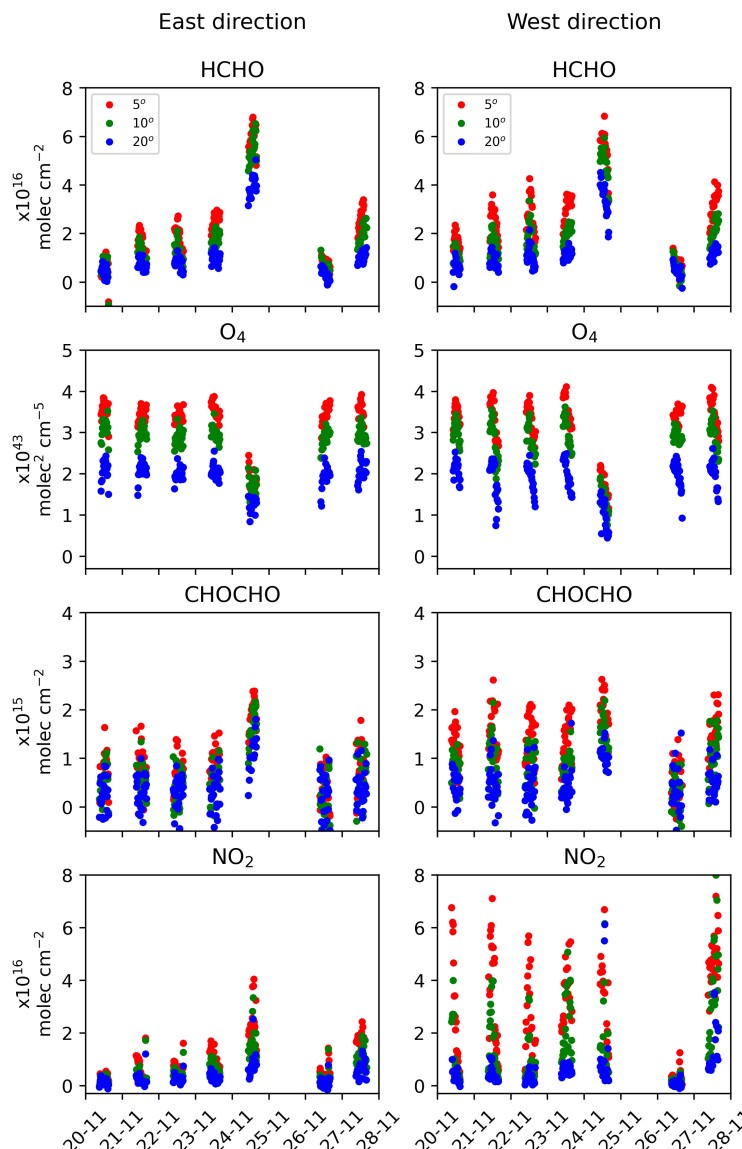

**Figure 6.** Differential Slant Column Densities for elevation angles 5° (red), 10° (green) and 20° (blue), for the trace gases HCHO, $O_4$, CHOCHO and $NO_2$ (first to fourth row, respectively) for the study period. Results for east and west directions are separated in the left and right panels, respectively. An increase in the dSCD values of HCHO is observed on November 24. The $O_4$ dSCD values observed for that date are lower and coalesce for the different elevation angles, which indicates a high aerosol load. The glyoxal signal also increases on November 24, although not as clearly as that of formaldehyde. The observed changes in $NO_2$ dSCD values for that day with respect to background are relatively small and may not be attributed to the event.

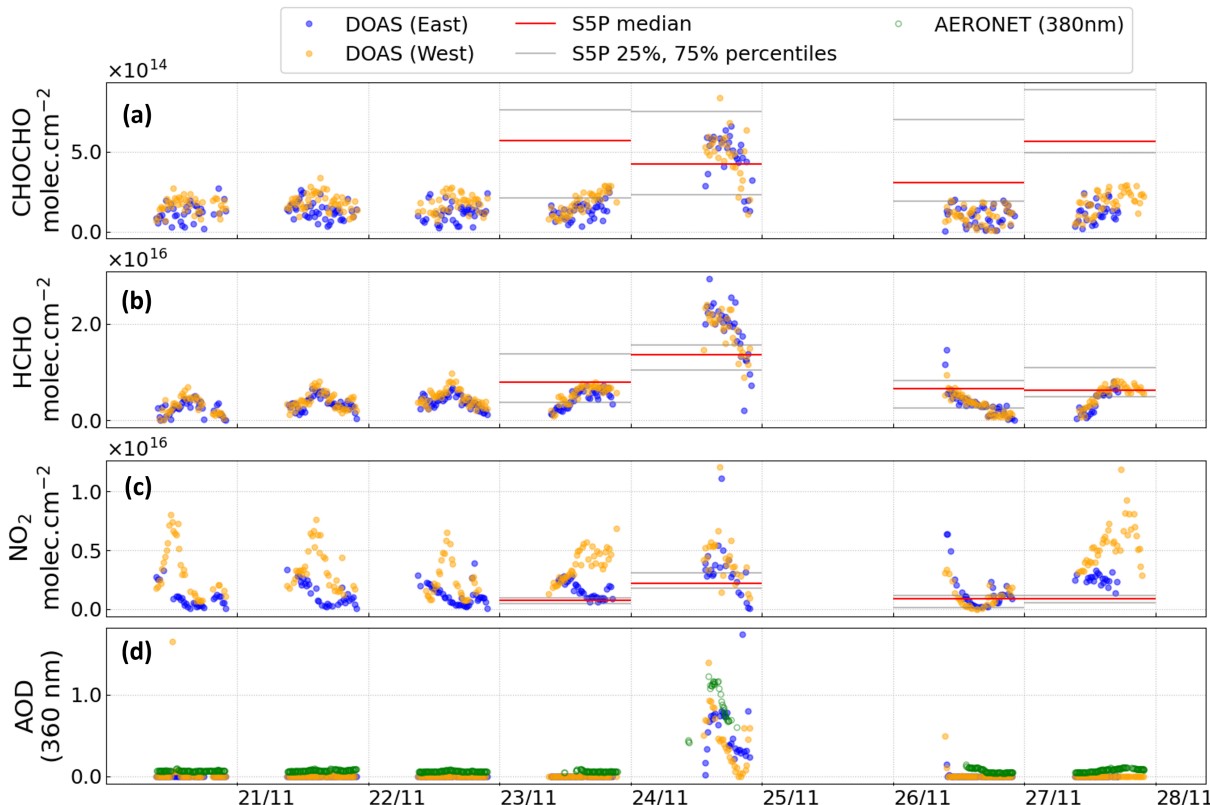

**Figure 7.** Panels (a-c) show the time series of Vertical Column Densities of glyoxal, formaldehyde and nitrogen dioxide. Percentile values for the corresponding products from TROPOMI-S5P over Montevideo are also shown as horizontal lines. Panel (d) shows the time series for the retrieved AOD by BOREAS from the MAX-DOAS observations and AERONET AOD values (at 360 and 380 nm, respectively).

## 4.2 Vertical Column Densities and profiles

We used the dSCD to derive the VCD with the BOREAS profile retrieval algorithm (Bösch et al., 2018). The VCD are related to the dSCD via the air mass factor (AMF), which depends on the local time, surface albedo, aerosol profile, trace-gas profiles, and the observation geometry (Section 2.6). The BOREAS output was screened to keep points with solar zenith angle less than 85 degress and with relative azimuth angle greater than 20 degrees.

Figure 7 shows the retrieved VCD of the trace gases and AOD considered in this study for the given period. We calculated percentile values for the corresponding TROPOMI-S5P products in a circular region with a radius of 25 km centered on Montevideo. These are shown as horizontal lines in panels a, b, and c. AERONET values for the same period are also displayed in panel (d). Figure 8(a) shows the AOD on November 24, comparing the BOREAS retrieval with AERONET values (at 360 and 380 nm, respectively), and Figure 8(b) shows the scatter plot and a linear fit between AERONET and BOREAS AOD values. The following remarks can be made regarding these results:

– The vertical column densities obtained from the BOREAS retrieval show a pronounced increase for formaldehyde and glyoxal on November 24 compared to the values of the other days, which is consistent with the passage of the plume over Uruguay found in satellite images (see Figures 2 and 3). The variability in the median values of TROPOMI-S5P CHOCHO, HCHO and $NO_2$ generally aligns with the corresponding MAX-DOAS measurements for most days compared. However, we note that there is a considerable variability in CHOCHO values – similar to MAX-DOAS observations – likely arising from the inherent difficulties in its retrieval.

– On this particular day, the AOD values derived from BOREAS are also above those from the days before and after November 24, qualitatively following the behaviour of the AERONET values (see Figure 7d). However, the latter are approximately 30% higher, as shown in Figures 8 (a) and (b). This discrepancy could be attributed to one of the following reasons: 1) an inhomogeneous spatial distribution of the aerosols within the plume, potentially causing a mismatch between MAX-DOAS and AERONET measurements which are taken in different directions; 2) the use of an a-priori profile during the inversion procedure that does not completely match the atmospheric conditions for November 24. Previous studies have suggested that scaling factors for the $O_4$ dSCD are sometimes needed to solve this issue (Wang et al., 2016; Wagner et al., 2019). In (Tirpitz et al., 2021), a similar behavior was detected during the second Cabauw Intercomparison of Nitrogen Dioxide Measuring Instruments campaign. However, the atmospheric conditions there were rather different from the ones present over Montevideo.

– In this study, only aerosols up to 4 km above mean sea level were considered in the BOREAS inversion model. It is possible that aerosols above the highest retrieval altitude or in altitudes with lower sensitivity were retrieved by the AERONET station but could not be retrieved by BOREAS. (Wang et al., 2016), reports a similar behavior in their study.

On November 24, the VCD values of formaldehyde increased by approximately a factor of four in comparison to the previous days, further supporting the conclusion that part of a plume, caused by a biomass burning event, was detected over Montevideo. No unusual activities were identified during that week in Montevideo. Atypical $NO_2$ values are not apparent for that day, compared to previous days. However, there is a good agreement when comparing VCD values derived from both viewing directions, which is unusual because of the difference in $NO_x$ emitters in the two directions. This could be explained by the fact that the transported plume covered all of Montevideo on that day. For days prior to the detection, the $NO_2$ diurnal cycle can be seen with its characteristic peak near noon, and the signal is stronger in the values computed from the west viewing direction, coinciding with the main industrial activity of the city such as the harbour and the oil refinery.

Glyoxal VCD values show an increase on November 24. This suggests the transport of glyoxal by a plume of distant origin. However, caution must be taken because the DOAS fitting error for glyoxal is relatively high for the period under study. The reason is that the CHOCHO dSCD is close to the detection limit, since Montevideo does not have strong sources of glyoxal. Hence, the majority of the glyoxal signal would likely originate from the transported plume.

Figure 9 shows examples of the retrieved gas and aerosol profiles for November 22 (top row), as reference, and for November 24 (bottom row).

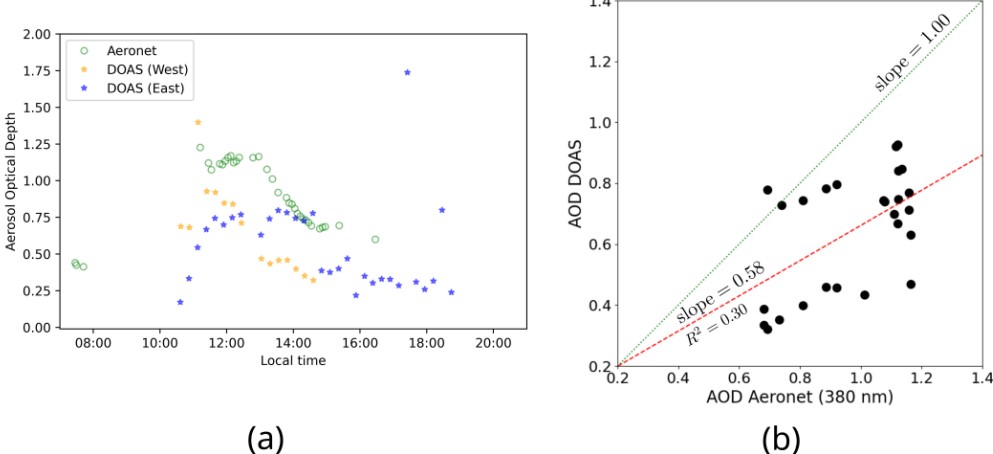

**Figure 8.** (a) AOD values obtained by AERONET at 380 nm and retrieved by BOREAS at 360 nm for November 24 in Montevideo. The latter is based on the observed $O_4$ dSCD for western and eastern viewing directions using MAX-DOAS. (b) Scatter plot and linear fit between AERONET and BOREAS AOD values for November 24.

The profiles show a change in the concentration and aerosol extinction during later hours of November 24. Glyoxal and formaldehyde show maxima in their profiles at approximately the same altitude, ranging between 1 and 2 km, which is consistent with the results obtained by the HYSPLIT model and supports our initial hypothesis, suggesting that gases and aerosols are transported at a similar altitude.

The aerosol profiles show a clean air situation on November 22, with the highest concentration close to the surface, indicating emissions primarily at or near the surface. On November 24, the aerosol profiles confirm the presence of a transported plume, characterized by atypically high aerosol loads in elevated layers over Montevideo. However, it is worth noting that accurately retrieving the upper boundary of the aerosol is not possible due to the lower sensitivity of MAX-DOAS inversion algorithms for higher altitudes, especially in the presence of high aerosol loads. This issue is more pronounced for aerosols, primarily due

to the non-linearity of the dependence of extinction to $O_4$ dSCD compared to trace gases.

### 4.3   Satellite observations of glyoxal and formaldehyde from TROPOMI

In order to evaluate the event on a larger scale, the products derived from the TROPOMI instrument were analyzed for the days around November 24. Figure 10 shows the different products considered for this analysis. On November 23, TROPOMI detects the beginning of a significant burning event near the south of Paraguay, which can be associated to various fire sources. Strong

signals of $CO$ and $NO_2$ result from incomplete biomass combustion and the result of oxidation processes. The UV aerosol index also shows high values in northern Argentina, associated with the movement of the plume to the south. Formaldehyde and glyoxal are detected near the origin of the plume. Glyoxal appears to have a less dispersed plume, most likely due to direct emission of glyoxal from fires, but there is also a contribution from oxidation processes within the plume. Formaldehyde

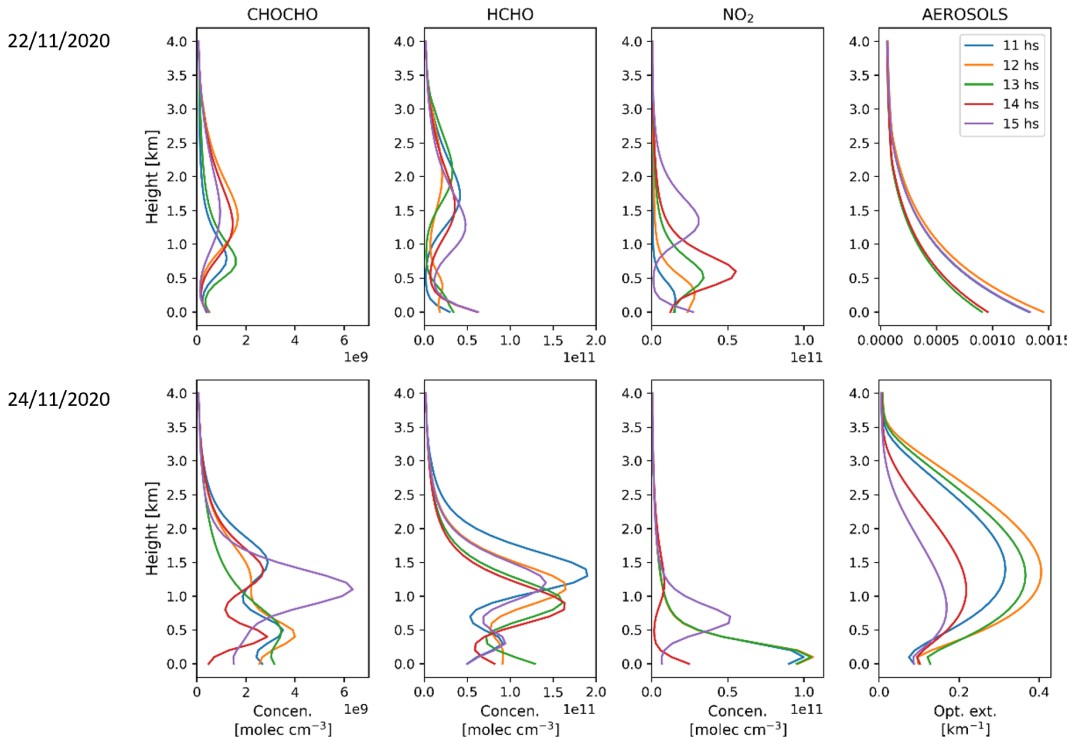

**Figure 9.** Retrieved gas and aerosol profiles for November 22 (top row) and November 24 (bottom row) at different hours during the day for the eastern direction.

is mainly produced through atmospheric oxidation processes. In TROPOMI data, glyoxal is less visible than formaldehyde
because the VCD is closer to the detection limit.

On November 24, new fire sources became active in the same region, leading to an increase of the size of the detected plume. Observations of the UV aerosol index and CO show the plume in the region of northern Argentina and southern Brazil, along with several $NO_2$ hotspots. These signals are attributed to emissions from the active fires since there are no other identifiable sources in this region, such as large cities, capable of producing these signals. Formaldehyde and glyoxal levels also show
an increase compared to the previous day, mainly due to the mixing with emissions from the new fire sources. The strongest signals of glyoxal and formaldehyde correlate with the RGB images of a dense plume within the Argentinian territory (see Figure 3). This can be associated to the well-known fact that glyoxal is formed in the early stages of a wildfire, influenced by factors such as the intensity of the fire, its extension and the number of ignition sources (Alvarado et al., 2020; Lerot et al., 2023) and references therein).
On this particular day, the aerosol index and CO columns, RGB images and the trajectories simulated with HYSPLIT, suggest that the plume moved across Uruguayan territory in southeast direction. However, there are no clear indications that the plume extends over the city of Montevideo in the TROPOMI observations of the short-lived species, as noted with the

percentile lines in Figure 7. The formaldehyde and glyoxal values are near to the detection limit, making it challenging to infer the overall pattern of the plume. Consequently, there are no significant changes compared to previous days. This could also be in part the result of cloudiness and the substantial aerosol load in the atmosphere over Montevideo, since aerosols are not accounted for in the glyoxal and formaldehyde retrievals (see Section 2.4). Typical mechanisms for the removal of both gases from the troposphere, such as the reaction with the OH radical and photolysis, may explain why the gases are not transported so far from the source (Atkinson, 2000). The disagreement between satellite and ground-based measurements over Montevideo is probably the result of the much better detection limit of the MAX-DOAS observations. It is also possible that the plume had not yet arrived at its highest concentrations at the satellite overpass time around noon.

A $NO_2$ spot over Montevideo is simply showing the typical pollution of the city itself. $NO_2$ produced by the biomass burning event is rapidly removed from the troposphere most likely through conversion to $HNO_3$. The evidence that part of the transport of the initial emissions is what is detected by the ground-based instruments is mostly provided by the CO columns product used as a tracer of biomass burning (Shi et al., 2015), since a small part of the border of the plume can effectively be seen over Montevideo (Figure 10).

On November 25, the Uruguayan territory experienced cloudy conditions, resulting in a gap in both satellite and terrestrial data. Despite the continued presence of sources in Paraguay and Argentina in the RGB images, their potential effect on Montevideo could not be detected after November 25.

## 5 Conclusions

A plume of aerosols and trace gases from biomass burning in the Paraguayan-Argentinian border region on November 22, 2020 was detected over Montevideo two days later. On November 24, AERONET measured very high AOD values compared to the usual Montevideo background in all bands. For example, there were AOD values at 380 nm greater than 1.0. Furthermore, ground-based MAX-DOAS measurements for that period showed an increase in formaldehyde and glyoxal vertical column densities, accompanied with an increase of the aerosol optical depth derived from the $O_4$ observations. Formaldehyde is known to be a useful tracer for biomass burnings. The observed increase of glyoxal levels was not so pronounced, probably due to the location of the ground-based instrument relative to the plume and the magnitude of the fires, so it is not clear that this gas can be used for plume detection in this scenario.

To confirm that the ground-based instruments detected this biomass burning plume, we used additional tools to ensure that the transport of the air masses to Uruguay was indeed possible. Examination of satellite RGB images shows the plume traveling from its source towards the vicinity of the city. Satellite data from TROPOMI show a vast carbon monoxide plume covering the entire south-east of Uruguay on November 24. The formaldehyde, nitrogen dioxide and glyoxal signals are relatively weak and not conclusive. Cloud coverage on November 25 prevented further investigation.

Trajectory simulations we conducted using HYSPLIT also support the conclusion that the plume reached Uruguay at the expected time. The air parcels are shown to be transported from the fire sources to the region near Montevideo, reaching a

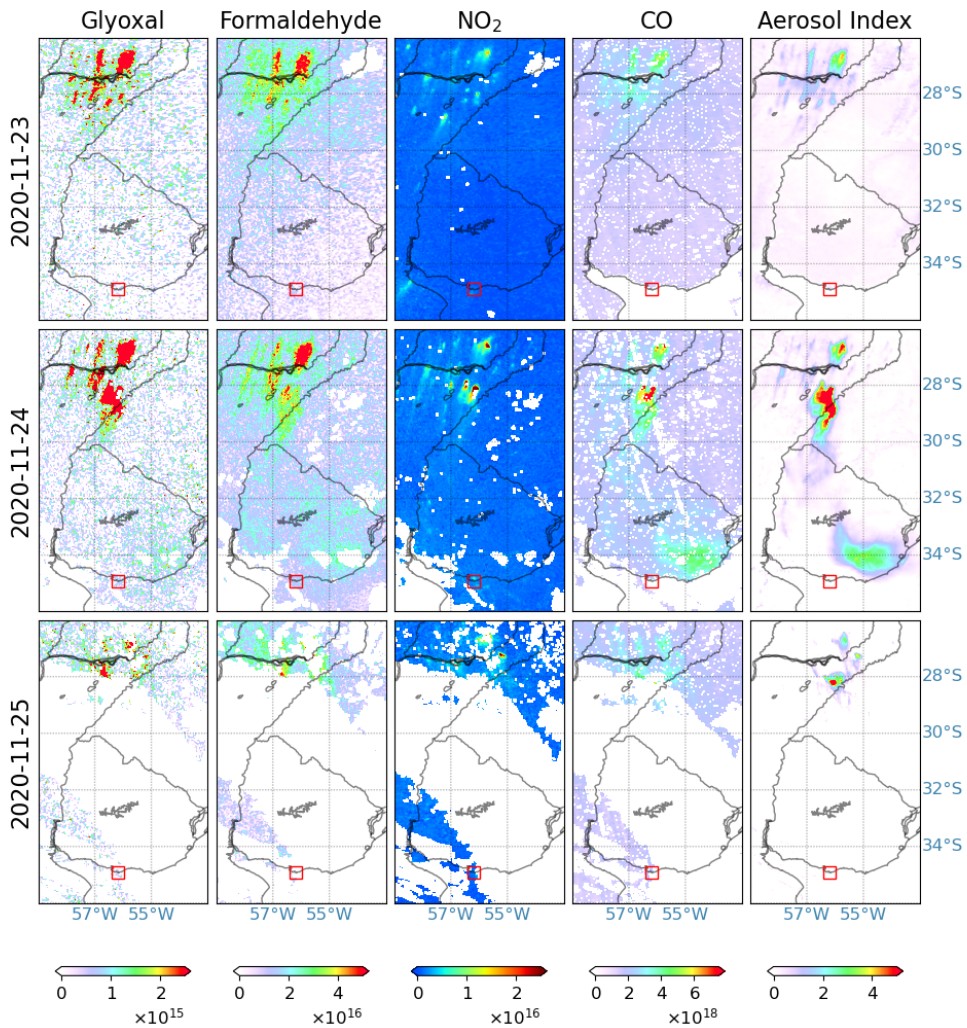

**Figure 10.** Vertical column densities (in $molec.\,cm^{-2}$) for glyoxal, formaldehyde, nitrogen dioxide and carbon monoxide, and the aerosol index (in arbitrary units) from TROPOMI on November 23, 24 and 25. The location of Montevideo is indicated by the red squares in the maps.

height of about 1.5 km above the ground, even when variations on the initial conditions are introduced. This agrees well with vertical profiles retrieved from the MAX-DOAS data using the BOREAS inversion algorithm for all species.

Atypical high trace gas concentrations and aerosol extinction levels confirm the ability of detecting transported plumes with ground-based instruments. The comparison of different trace gas and aerosol profiles shows possible temporal and spatial inhomogeneities within the plume for different species.

The above points can be interpreted as a partial validation of the ground-based detection of the plume, and we think that, in the future, ground-based atmospheric monitoring in Montevideo can be used as a tool for the detection of biomass burning events on its own, especially when satellite data is inconclusive for certain species such as HCHO, $NO_2$ and CHOCHO.

  As shown in this case-study, ground-based remote sensing measurements of trace gases and aerosols are useful tools to detect the impact of biomass burning emissions at long distances from the source. Such measurements are much more sensitive

than satellite data and therefore complement space-borne observations which provide full spatial coverage. Continuous measurements of this type of stations at several locations on the South-American continent would provide valuable information on the regional and continental scale effects of wild fires. They also would deliver the data needed to validate atmospheric models to investigate the chemical processes in the plumes.

*Data availability.* AERONET data for Montevideo_FING station can be found at https://aeronet.gsfc.nasa.gov/cgi-bin/data_display_aod_

v3?site=Montevideo_FING&nachal=2&level=2&place_code=10. BOREAS data for the days involved in this study are available at https://www.fing.edu.uy/if/grupos/optica_aplicada/assets/ (Casaballe, 2023). TROPOMI data are available upon request from Leonardo Alvarado (leonardo.alvarado@awi.de). ICESat-2 data can be found at EARTHDATA-NASA database at https://search.earthdata.nasa.gov/search?q=ATL09%20V006.

*Code and data availability.* QDOAS software can be download from Royal Belgian Institute for Space Aeronomy webpage (https://uv-vis.

aeronomie.be/software/QDOAS/) (Danckaert et al., 2017). Access to BOREAS software can be requested to the Institute of Environmental Physics, University of Bremen. Access to run the HYSPLIT trajectory model can be found in https://www.ready.noaa.gov/HYSPLIT_traj.php

## Appendix A: ICESat-2 observations of the biomass burning event

Data from the Ice, Cloud and land Elevation Satellite-2 (ICESat-2) was analysed searching for additional information about the BB plume passing over Uruguay. ICESat-2 is part of the Earth Observing System from NASA, and its main mission is to

measure ice sheet and glaciers elevation, as well as to measure and study ice thickness and vegetation characteristics in forests worldwide. It is equipped with the Advanced Topographic Laser Altimeter System (ATLAS) instrument which times the travel of 532 nm laser pulses for high resolution altimetry to perform surface and atmospheric measurements (Palm et al., 2021).

  The data product analysed in this study was ATL09 Level 3A, which contains calibrated attenuated backscatter profiles and layer integrated attenuated backscatter from the data acquired by ATLAS (Palm et al., 2022).

Between November 23 and 24, the satellite performed two overpasses over Uruguay with a time stamp close to the event and with a trajectory almost over the BB plume. Figures A1 and A2 show the layer data for November 23 and 24 respectively. Only cloud and aerosol layers are detected over Uruguay in these passes.

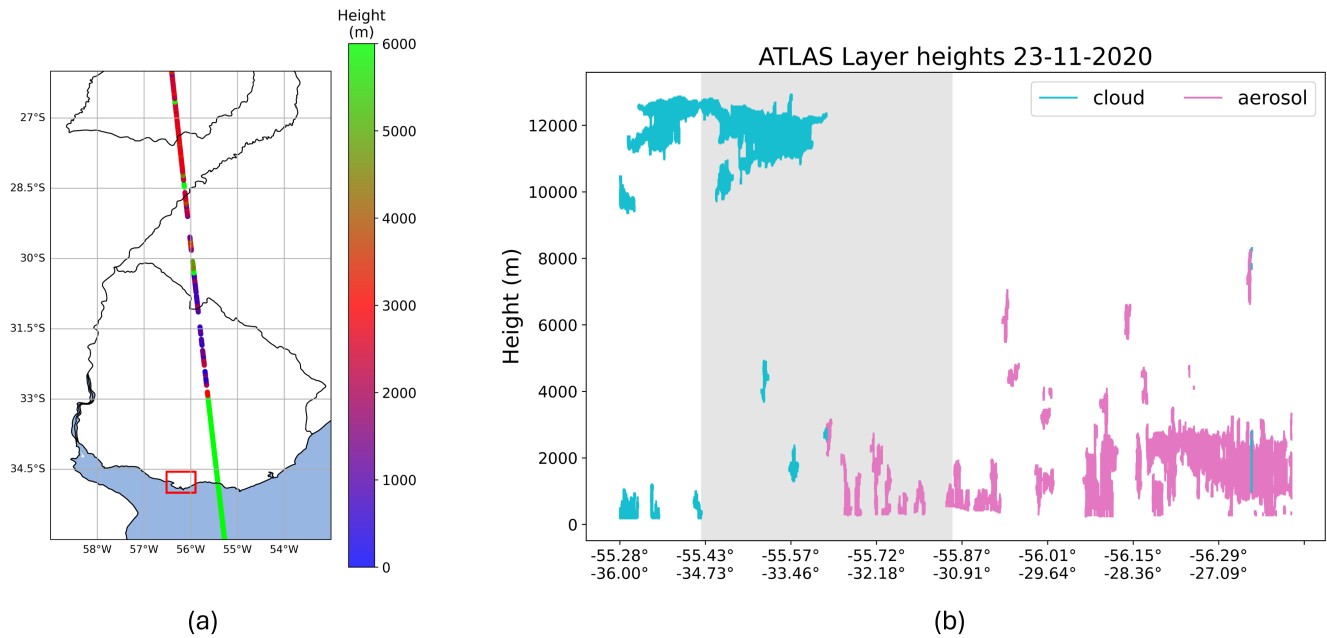

**Figure A1.** (a) ICESAT-2 ascending track above Uruguay for November 23, 2020. The color scale represents the height of the topmost layer detected. The red square indicates the location of Montevideo. (b) Distribution of clouds an aerosol layers along the trajectory, from ATLAS ATL09 product data.

Figure A1 reveals that the plume has not yet reached Uruguayan territory at the time of the overpass on November 23, as no aerosol layers were detected. A low-altitude aerosol signal appears at northern latitudes when the satellite passes over the regions of Argentina and Paraguay, close to the location of the fires. However, it was not possible to obtain independent confirmation with additional satellite imagery (such as the VIIRS instrument) since the ICESat-2 overpass occurs at nighttime.

For November 24, an aerosol layer was detected in the eastern part of Uruguay, approximately 200 to 375 kilometers from Montevideo, at heights mostly between 1500 and 4000 m, as shown in Figure A2. This result is in agreement with the profiles obtained from the BOREAS inversion analysis and with the VIIRS RGB images (see Figure 3).

*Author contributions.* The project was conceptualised by EF and LMAA. The analysis and discussion of results were carried out by MO, EF, AA, TB, NC, AR and LMAA. The original draft was written by MO. All authors contributed to the final manuscript.

*Competing interests.* Andreas Richter is Editor of the journal.

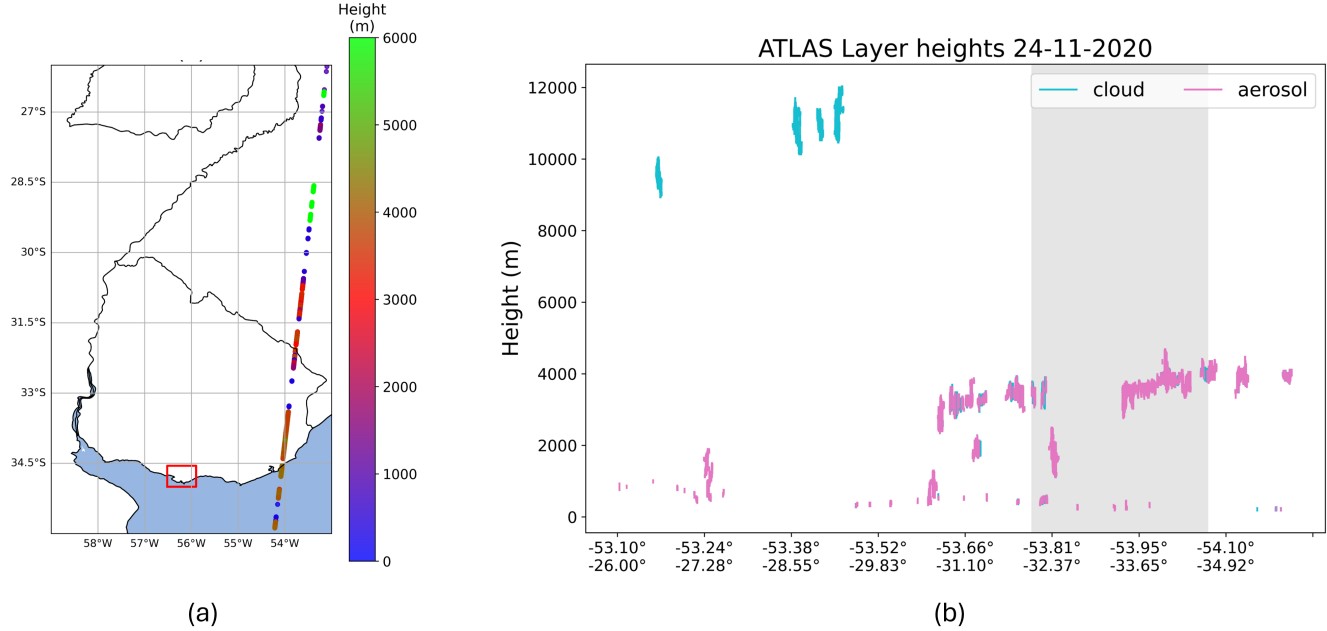

**Figure A2.** (a) ICESAT-2 descending track above Uruguay for November 24, 2020. The color scale represents the height of the topmost layer detected. The red square indicates the location of Montevideo. (b) Distribution of clouds an aerosol layers along the trajectory, from ATLAS ATL09 product data.

*Acknowledgements.* AA acknowledges the Agencia Nacional de Investigación e Innovación (ANII) for the funding POS_NAC_2022_1_174198. MO acknowledges Comisión Académica de Posgrado for the PhD fellowship support. We acknowledge the use of imagery provided by services from NASA's Global Imagery Browse Services (GIBS), part of NASA's Earth Observing System Data and Information System (EOSDIS) and the National Snow and Ice Data Center (NSIDC) for the access to the ICESat-2 data. We thank the TROPOMI team for providing satellite products. Copernicus Sentinel-5P level-1 data and level-2 $NO_2$, $CO$, and UV Aerosol Index data were used in this study. This publication contains modified COPERNICUS Sentinel data. We gratefully acknowledge the NOAA Air Resources Laboratory (ARL) for the provision of the HYSPLIT transport and dispersion model and/or READY website (https://www.ready.noaa.gov) used in this publication. EF thanks Brent Holben (NASA Goddard Space Flight Center) for allowing the Applied Optics Group to participate in the AERONET project. The authors thank the anonymous reviewers for their valuable and constructive comments that helped to improve the manuscript.

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
