# Peer review of "Measurement Report: Combined use of MAX-DOAS and AERONET ground-based measurements in Montevideo, Uruguay for the detection of distant biomass burning"

_EGUsphere, 2023_

## Referee Comment (RC3)

Review of manuscript 'Measurement Report: Potential of MAX-DOAS and AERONET ground based measurements in Montevideo, Uruguay for the detection of distant biomass burning' by Osorio et al, submitted for publication in Egusphere.

General review

The paper describes the detection of a biomass burning generated plume, consisting of a mixture of aerosol particles and gaseous components, using ground-based observations of UV-VIS scattered radiation by a ground-based multi-axis differential optical absorption spectrometer (MAX-DOAS) in Montevideo, Uruguay during November 2020. The authors made use of collocated AERONET observations on total column aerosol optical depth, particle size distribution, and aerosol single scattering albedo in the inversion MAX-DOAS spectroscopic observations to derive the vertical distributions of formaldehyde (HCHO), glyoxal (CHOCHO), nitrogen dioxide ($NO_2$), oxygen collision complex ($O_4$), and aerosols.

The obtained results show that the combined use of MAX-DOAS and AERONET observation can be advantageously used to derive the vertical distribution of transported carbon-containing aerosols generated by biomass burning. Although the authors make a convincing description of their results, a few, mostly minor, modifications are needed before the manuscript is acceptable for publication.

Specific comments

Title: The current title does not adequately convey the main contribution of this work. I suggest slightly modifying the tittle to: 'Combined use of MAX-DOAS and AERONET ground-based observations in Montevideo, Uruguay, for retrieving the vertical distribution of biomass burning generated aerosols.'

Abstract: Since the abstract is a standalone description of the paper, all acronyms therein must be resolved.

Line 5 Resolve MAX-DOAS acronym

Line 7 Resolve AERONET acronym

Line 8 Resolve AOD acronym

Line 11 Resolve TROPOMI and HYSPLIT acronyms

All acronyms must be resolved again in the main body of the manuscript.

Line 29 Resolve BB acronym

Line 38 Provide a reference for the statement on air quality in Uruguay.

Line 40 Suggest using 'sunphotometer' to be consistent with AERONET's terminology.

Line 54-57 The statement on the variability of AERONET AOD and other parameters, and apparent consistency with the detection of 'this plume' should be supported with observations. Likewise, conclusions on satellite data analysis and trajectory simulations do not belong in the introduction section. At this point in the paper, these statements read like unsupported speculations.

Line 73. What is the resulting vertical resolution?

Line 85. Add a reference on AERONET current version and aerosol data products.

Line 90. Resolve UV-VIS-NIR-SWIR acronyms.

Line 94 Resolve CCD acronym.

Line 100 Resolve VOC acronym

Line 106 Resolve TM5-MP and SCD acronyms.

Line 125. Add references on retrievability of aerosol vertical distribution by this technique.

Line 126. List specific AERONET aerosol retrieval results extrapolated to the matching windows and discuss the extrapolation method.

-Which AERONET wavelengths were used in the calculation of Angstrom Exponent?

- For this application, AERONET shortest wavelengths (340 and 380 nm) should have been used.

- AERONET does not retrieve single scattering albedo at wavelengths shorter than 440 nm. How was aerosol absorption accounted for in the UV region?

Line 160. CALIPSO or icesat-2 lidar data may have been available on this day. If so, how does it compare?

Line 173. Can these results be confirmed with CALIPSO or icesat2 lidar observations? CALIPSO data may be affected by the South Atlantic Anomaly. Icesat2, however, reports observations over the analysis period in this paper at https://icesat-2.gsfc.nasa.gov/atmo-data.

Line 175. Discuss the expected lifetimes of the retrieved species. It may be relevant in this analysis.

Line 182. Resolve QDOAS acronym.

Line 182. Discuss Tables 1 and 2 separately. They provide different information. Describe each column of Table 1 in detail. No absorption cross-section data is shown in Table 2.

Line 189 The description of Figure 5 is incomplete. Only the top row is discussed. There is not an adequate discussion of the bottom three rows. The figure caption should accurately describe what is shown in each of the nine panels of the figure. If some of the data shown in this figure is not relevant to the analysis, then remove it.

Line 193. Describe in some detail the mentioned color index classification algorithm. How is the comparison of intensities used to discern clear from cloudy or partly cloudy conditions?

Line 198. Change the x-axis time representation in Figure 6 to the commonly used month-day nomenclature. The figure caption should clearly indicate what is shown on each row from top to bottom, and on each column from left to right.

Line 219. Sentinel 5 (S5P) is just the satellite name. The sensor's name should also be included.

Line 225. Change the x-axis time representation in Figure 7 to the commonly used month-day nomenclature.

Line 227. AERONET does retrieve AOD at 360 nm. Clarify in the text, and in the Figure 7 caption the wavelength of the AERONET AOD retrieval. The same comment applies to the description of AERONET AOD shown in Figure 8a.

Lines 230-232. This statement makes no sense. AERONET measurements are insensitive to formaldehyde and glyoxal. Please rephrase.

Line 243 Resolve CINDI-II acronym.

Line 245. CALIPSO lidar data, if available, should confirm the presence (or not) of aerosols above 4 km.

Line 277. The authors seem to refer to the UV Aerosol Index (UVAI). Since it is the name of a specific satellite product it should be capitalized. The authors should briefly explain why the UVAI readings are relevant in the context of this work and provide relevant references. In addition to the qualitative UVAI product, 388 nm aerosol optical depth and single scattering albedo are also available from TROPOMI observations (Torres et al., AMT, 2020).

Line 283 UV Aerosol Index

Line 291 UV Aerosol index

Line 312 Include the corresponding AERONET wavelength.

---

## Author Comment (AC1)

**We gratefully thank the reviewers for the constructive comments and suggestions to improve the manuscript. Below are the detailed responses to their comments. The reviewers' comments are listed black in italics and our responses and changes in the manuscript are shown in blue. The changes in the revised manuscript are also highlighted.**

**Referee #1 Report**

*General comment:*

*The manuscript investigates with, ground based and satellite measurements, the effects on the air composition over the Montevideo region during a long range transport due to a biomass burning event. Investigations of air quality and the consequent effects a wildfire can cause on the environment are very important to properly address. Detecting these events and following their trajectories are the initial steps in building a scientific understanding of these processes. The authors showed good results in demonstrating the gases/particles contributions from the wildfire event. My major concern is with the following investigation the authors could have done with the elements showed in the paper. For example, the vegetation damage aspects or the fire aging plume characteristics and its chemical/physic composition and interaction with the radiation could have been explored.*

**We understand the potential for a broader investigation than the one we present, including other aspects that Referee #1 highlighted through the General and Specific comments. However, the scope and objectives of our research focus on detection using mainly DOAS and aerosol measurements (especially the comparison between ground-based and satellite measurements). While such combination of measurements is common on the northern hemisphere, they are rare in South America. This type of events could affect the atmosphere over Uruguay and its surroundings and there is almost no information about it. We are presenting, to the best of our knowledge, a first case of detection over South America with these methods, emphasizing the understanding of the trajectory of the wildfire plume and its detection over Montevideo.**

*Specific comments:*

*The authors should improve the investigation by analyzing the amount of O3 produced due to the fire event and how it might cause damage to the atmosphere and environment, for example vegetation damage.*

*Another aspect that could be further explored is the spatial distribution of the fire plume effect. Depending on the fire plume age, what are the secondary organic aerosol, Black carbon, AOD, CO, O3, NOx, HCHO values? And how are they impacted by the solar radiation for example, in the formation processes of O3 and SOA?*

*It is important to know during a fire event, how strong is the correspondence between atmospheric oxidants, such as, characterized O3 and NO2 and the SOA concentration.*

**The purpose of our work is to show the potential of terrestrial remote sensing that allows us to quantify some atmospheric constituents associated with the burning of biomass in large areas that affect Montevideo and its surroundings. Although the relationship between the characteristics of the fire and the dynamics of the emitted substances is a very important and interesting topic, we consider it to be outside the objectives of this study. A closer look at these aspects would have to rely heavily on chemical modelling which is best done in an independent work.**

*The authors could be using the Hysplit model to define forward trajectories by calculating it starting from a cluster of points at varying heights above sea level. For example, defining a cylinder of initial staring points with a radius of ~5km centered on the fire plume's initial location, the average gas and aerosol concentrations could be calculated within the volume defined in latitude and longitude by the points as they are time evolved by HYSPLIT, rather than just examine point values. The authors could also define an altitude range to track the fire plume.*

*It is important to have a numerical experiment with an atmospheric model with a gas-phase chemistry, and aerosol model to better understand the plume trajectory. Specifically when no ground or satellite data are available.*

**We understand that an exhaustive study that includes the trajectories and chemical modelling of the composition of the products generated during the emissions and transport could be an interesting contribution. However, this is beyond the scope of this work.**

**In this study, HYSPLIT is used as a complementary tool to confirm the origin of the burning sources, since satellite imagery has a lower temporal resolution than our ground-based instruments in Montevideo.**

*I recommend the authors to better emphasize the main scientific question approached in the paper. The contribution in this paper needs to be solid and be part of the knowledge's support used for future science.*

**We hope that the improvements made in the new version of the manuscript will make the main scientific questions become clearer, please see the new version.**

---

## Author Comment (AC2)

**We gratefully thank the reviewers for the constructive comments and suggestions to improve the manuscript. Below are the detailed responses to their comments. The reviewers' comments are listed in black italics and our responses and changes in the manuscript are shown in blue. The changes in the revised manuscript are also highlighted.**

**Referee #2 report**

*Review of*

*Measurement Report: Potential of MAX-DOAS and AERONET ground based measurements in Montevideo, Uruguay for the detection of distant biomass burning*

*by Matías Osorio et al*

*https://doi.org/10.5194/egusphere-2023-2390*

*Summary: This study reports observations during an episode of long range transport of smoke using a MAX-DOAS instrument located in Montevideo, Uruguay. The observations reported are complemented with ancillary standard information available such as Aeronet aerosol products, HYSPLIT model runs to trace back the source regions and satellite products from the TROPomi sensor. The novelty of this study is the first-time description of a smoke event in this region of the world with a combination of instrumentation that has become standard in the northern hemisphere but are notoriously lacking in the southern hemisphere. Studies like this one are very much welcomed. I view this study as a reference work where future research focusing on smoke transport in South America can compare with. This fact makes this study worthy of publication and I recommend to it. I do have general comments about writing style, and suggestions for figure improvement and clarifications. Not addressing them should not be a reason to prevent the publication however, it would make a more enjoyable reading and useful information for future studies as I detail below.*

**We appreciate the positive opinion of Referee #2 about the study and the manuscript. Please, find below the response to the detailed comments.**

*Detail:*

*1) While this is a measurement report and there is no expectation of new discovery reporting or lengthy discussions and speculation, the general tone of the paper is too tentative in the sense that it is written as if the observations must demonstrate that what it is been observed by the MAX-DOAS is smoke. I think the emphasis should be changed to a different tone where established techniques are used to track a smoke event. The goal is to report how these observations and analysis perform in a region/environment where there hasn't been many application of such techniques. In other words, it is not a matter to prove that smoke was arriving at Montevideo but the goal is to report how a smoke event is observed in Montevideo with these commonly used techniques in the northern Hemisphere but not in this region. For example expressions like:*

*lines 204-205: "The main interest in this analysis is to assess the effectiveness of formaldehyde and glyoxal as plume tracers from biomass burning events and their detection in Montevideo."*

*Unless there is an alternative reason to suspect that formaldehyde and glyoxal are coming from somewhere else and given that it already been seen in smoke events elsewhere, it is not necessary to be so tentative to prove the smoke is coming from where the satellite images and model clearly show. This tone pervades the paper and it distracts from the real novelty which is to highlight how such event is seen by a tool that is not common in the South America. So, I highly advice to do some style editing.*

**We followed this suggestion and the text in the new version of the manuscript was modified accordingly, with which we hope that the new version will better define the objective of this work.**

*2) With regards to readability the paper is perfectly understandable and fine. However, there are some mistakes particularly regarding the use of prepositions. I will not detail those because it is too tedious, but I advise to have it check by a native speaker.*

**We reviewed and corrected the new version of the manuscript, as suggested.**

*3) Figures: overall I found the figures clear and appropriate, except for couple of cases I mention below*

**Thank you for the positive comment and feedback.**

*4) line 225. Can you clarify how those percentiles are computed? they are not explained in the satellite section and as far as I know, there are not part of the standard satellite products.*

**We defined a region of interest as a circle of 25 km radius centered in Montevideo. Values over water and invalid data were discarded. We calculated percentiles 25th, 50th and 75th with the values of the pixels in the region of interest after the screening.**

**We added the following statement in the manuscript (from line 236):**

**"We calculated percentile values for the corresponding TROPOMI-S5P products in a circular region with a radius of 25 km centered in Montevideo. These are shown as horizontal lines in panels a, b, and c."**

*5) line 228-230 , figure 8b is a scattering plot for which there is linear fit plotted. It is not a correlation plot. Please correct.*

**Done.**

*6) Paragraph starting 236 , Not clear to what plot or figure this explanation is referring to , so it is difficult to verify.*

**This explanation refers to the results presented in Figures 7 and 8. To improve the text, we added this information to the start of the paragraph:**

**"On this particular day, the AOD values derived from BOREAS are also above those from the days before and after November 24, qualitatively following the behaviour of the AERONET values (see Figure 7(d)). However, the latter are approximately 30% higher, as shown in Figures 8 (a) and (b)."**

*7) full text in page 15 (starting line 248). This is a difficult description to follow because the data for Nov24 shown in figure 7 is too cramped. I think it would be more illustrative to add a figure displaying a time series of hourly observations for 22 and 24/Nov to illustrate this paragraph.*

**Aside from those two dates, in Figure 7 we aim to display VCD values in Montevideo that were not affected by the BB event. Therefore, we decided to maintain the extended period in Figure 7.**

*8) incidentally, I do not think that figure 9 is very useful because there is little independent information to verify the profiles. I think that replacing it with the above suggested new figure would be more useful.*

**We agree with the Referee's statement that independent information is not available, and that the verification of the shown profiles is therefore not possible. From a plausibility perspective, the spatial and temporal evolution of the species are nevertheless reasonable. We believe that figure 9 is worth keeping in the manuscript, as it is the only figure which shows the differences in altitude for the individual specie, as all other figures show integrated values only.**

**Figure 9 might also help future authors to compare with their results and to further analyse the different spatial inhomogeneities within larger plumes at other biomass burning events.**

---

## Author Comment (AC3)

**We gratefully thank the reviewers for the constructive comments and suggestions to improve the manuscript. Below are the detailed responses to their comments. The reviewers' comments are listed in black italics and our responses and changes in the manuscript are shown in blue. The changes in the revised manuscript are also highlighted.**

**Referee #3 report**

*Review of manuscript 'Measurement Report: Potential of MAX-DOAS and AERONET ground based measurements in Montevideo, Uruguay for the detection of distant biomass burning' by Osorio et al, submitted for publication in Egusphere.*

*General review*

*The paper describes the detection of a biomass burning generated plume, consisting of a mixture of aerosol particles and gaseous components, using ground-based observations of UV-VIS scattered radiation by a ground-based multi-axis differential optical absorption spectrometer (MAX-DOAS) in Montevideo, Uruguay during November 2020. The authors made use of collocated AERONET observations on total column aerosol optical depth, particle size distribution, and aerosol single scattering albedo in the inversion MAX-DOAS spectroscopic observations to derive the vertical distributions of formaldehyde (HCHO), glyoxal (CHOCHO), nitrogen dioxide (NO2), oxygen collision complex (O4), and aerosols.*

*The obtained results show that the combined use of MAX-DOAS and AERONET observation can be advantageously used to derive the vertical distribution of transported carbon-containing aerosols generated by biomass burning. Although the authors make a convincing description of their results, a few, mostly minor, modifications are needed before the manuscript is acceptable for publication.*

**We thank Referee #3 for the opinion about the manuscript and the suggestions made. In the following, the changes made in response to the Referee's suggestions are addressed.**

*Specific comments*

*Title: The current title does not adequately convey the main contribution of this work. I suggest slightly modifying the tittle to: 'Combined use of MAX-DOAS and AERONET ground-based observations in Montevideo, Uruguay, for retrieving the vertical distribution of biomass burning generated aerosols.'*

**We followed Referee suggestion and conciliate with the comments of Referee #2, to emphasize the novelty of the detection of this type of events in South America. Since the vertical distribution of the constituents is not the main result of this manuscript, we opted to change the title to "Combined use of MAX-DOAS and AERONET ground-based measurements in Montevideo, Uruguay for the detection of distant biomass burning".**

**We believe that with this change the title better describes the purpose of the article.**

*Abstract: Since the abstract is a standalone description of the paper, all acronyms therein must be resolved.*

**Done.**

*Line 5 Resolve MAX-DOAS acronym*

**Done.**

*Line 7 Resolve AERONET acronym*

> **Done.**

*Line 8 Resolve AOD acronym*

> **Done.**

*Line 11 Resolve TROPOMI and HYSPLIT acronyms*

> **Done.**

*All acronyms must be resolved again in the main body of the manuscript.*

*Line 29 Resolve BB acronym*

> **Done.**

*Line 38 Provide a reference for the statement on air quality in Uruguay.*

> **We added two references in the Introduction section related to public air quality reports generated by Uruguayan government and Montevideo city council (from line 38):**
>
> **"Air quality in Uruguay is generally good (Ministerio de Ambiente, 2024; Intendencia de Montevideo, 2024); however…"**
>
> Ministerio de Ambiente, República Oriental del Uruguay: Informes de monitoreo y documentos de calidad de aire, https://www.gub.uy/ministerio-ambiente/politicas-y-gestion/informes-monitoreo-documentos-calidad-aire, accessed: 28/02/2024, 2024.
>
> Intendencia de Montevideo: Calidad del aire, https://montevideo.gub.uy/calidad-del-aire, accessed: 28/02/2024, 2024.

*Line 40 Suggest using 'sunphotometer' to be consistent with AERONET's terminology.*

**Done.**

*Line 54-57 The statement on the variability of AERONET AOD and other parameters, and apparent consistency with the detection of 'this plume' should be supported with observations. Likewise, conclusions on satellite data analysis and trajectory simulations do not belong in the introduction section. At this point in the paper, these statements read like unsupported speculations.*

**We followed the suggestion and removed the lines 54-57 and replaced them with this new sentence:**

**"This relatively strong event resulted in the detection of a plume passing over Montevideo, by means of multiple instruments covering ground-based and satellite observations."**

*Line 73. What is the resulting vertical resolution?*

**We are describing the optical setup for our DOAS instrument in the text. We are using a common configuration with a small field of view (less than one degree) and multiple elevation angles, in steps of one or more degrees, scanning a vertical plane. For instance, with a 0.3 degree vertical field of view, the instrument covers an area of approximately 50 m at 10 km of distance.**

**The result of the measurements is the input for the inversion algorithm described in the BOREAS section. The vertical resolution of the retrieved profiles depends on the information content of the measurements, the altitude, the solar position and the aerosol load in the atmosphere. It typically increases from a few tens of meters at the surface to several kilometers in the middle troposphere (see also the references in the answer to the comment regarding line 125 below).**

*Line 85. Add a reference on AERONET current version and aerosol data products.*

**We added the following references regarding the Referee comment:**

Dubovik, O., & King, M. D. (2000). A flexible inversion algorithm for retrieval of aerosol optical properties from Sun and sky radiance measurements. Journal of Geophysical Research: Atmospheres, 105(D16), 20673–20696. https://doi.org/10.1029/2000JD900282

Dubovik, O., Smirnov, A., Holben, B. N., King, M. D., Kaufman, Y. J., Eck, T. F., & Slutsker, I. (2000). Accuracy assessments of aerosol optical properties retrieved from Aerosol Robotic Network (AERONET) Sun and sky radiance measurements. Journal of Geophysical Research: Atmospheres, 105(D8), 9791–9806. https://doi.org/10.1029/2000JD900040

Holben, B. N., Tanré, D., Smirnov, A., Eck, T. F., Slutsker, I., Abuhassan, N., Newcomb, W. W., Schafer, J. S., Chatenet, B., Lavenu, F., Kaufman, Y. J., Castle, J. vande, Setzer, A., Markham, B., Clark, D., Frouin, R., Halthore, R., Karneli, A., O'Neill, N. T., … Zibordi, G. (2001). An emerging ground-based aerosol climatology: Aerosol optical depth

from AERONET. Journal of Geophysical Research: Atmospheres, 106(D11), 12067–12097. https://doi.org/10.1029/2001JD900014

Sinyuk, A., Holben, B. N., Eck, T. F., Giles, D. M., Slutsker, I., Korkin, S., Schafer, J. S., Smirnov, A., Sorokin, M., & Lyapustin, A. (2020). The AERONET Version 3 aerosol retrieval algorithm, associated uncertainties and comparisons to Version 2. Atmospheric Measurement Techniques, 13(6), 3375–3411. https://doi.org/10.5194/amt-13-3375-2020

**We rephrased the paragraph to improve readability:**

**"The AERONET node at Montevideo uses a CIMEL CE3128-T multispectral sunphotometer (Holben et al., 1998) installed adjacent to the MAX-DOAS instrument. This sunphotometer measures direct and diffuse solar radiation at different bands ranging from 340 to 1640 nm to retrieve aerosol data from the observations. AERONET provides several data products derived from these measurements, e.g. aerosol optical depth (AOD), single scattering albedo (SSA) and phase function (Holben et al., 2001; Dubovik et al., 2000; Dubovik and King, 2000). In our study we used AERONET V3 level 2.0 data, ensuring good quality of data by applying radiometric and instrumental corrections, as well as cloud-cover removal algorithms (Sinyuk et al., 2020)."**

*Line 90. Resolve UV-VIS-NIR-SWIR acronyms.*

**Done.**

*Line 94 Resolve CCD acronym.*

**Done.**

*Line 100 Resolve VOC acronym*

**Done.**

*Line 106 Resolve TM5-MP and SCD acronyms.*

**Done.**

*Line 125. Add references on retrievability of aerosol vertical distribution by this technique.*

**The retrievability and accuracy of aerosol profiles retrieved by MAX-DOAS inversion algorithms is discussed in e.g. Bösch et al. (2018), Frieß et al (2019) and Tirpitz (2021). In general, the vertical resolution of MAX-DOAS inversion algorithms is the highest close to the surface and decreases with altitude. Close to the surface, a vertical resolution of roughly 100m is possible but it decreases to more than 500m within the first kilometer (Bösch et al. 2018).**

Bosch, T., Rozanov, V., Richter, A., Peters, E., Rozanov, A., Wittrock, F., Merlaud, A., Lampel, J., Schmitt, S., de Haij, M., Berkhout, S., Henzing, B., Apituley, A., den Hoed, M., Vonk, J., Tiefengraber, M., Muller, M., and Burrows, J. P.: BOREAS – a new MAX-DOAS profile retrieval algorithm for aerosols and trace gases, Atmospheric Measurement Techniques, 11, 6833–6859, https://doi.org/10.5194/amt-11-6833-2018, 2018.

Fries, U., Beirle, S., Alvarado Bonilla, L., Bosch, T., Friedrich, M. M., Hendrick, F., Piters, A., Richter, A., van Roozendael, M., Rozanov, V. V., Spinei, E., Tirpitz, J.-L., Vlemmix, T., Wagner, T., and Wang, Y.: Intercomparison of MAX-DOAS vertical profile retrieval algorithms: studies using synthetic data, Atmospheric Measurement Techniques, 12, 2155–2181, https://doi.org/10.5194/amt-12-2155-2019, 2019.

Tirpitz, J. L., Fries, U., Hendrick, F., Alberti, C., Allaart, M., Apituley, A., Bais, A., Beirle, S., Berkhout, S., Bognar, K., Bosch, T., Bruchkouski, I., Cede, A., Chan, K. L., Hoed, M. D., Donner, S., Drosoglou, T., Fayt, C., Friedrich, M. M., Frumau, A., Gast, L., Gielen, C.,Gomez-Martin, L., Hao, N., Hensen, A., Henzing, B., Hermans, C., Jin, J., Kreher, K., Kuhn, J., Lampel, J., Li, A., Liu, C., Liu, H., Ma, J., Merlaud, A., Peters, E., Pinardi, G., Piters, A., Platt, U., Puentedura, O., Richter, A., Schmitt, S., Spinei, E., Zweers, D. S., Strong, K., Swart, D., Tack, F., Tiefengraber, M., Hoff, R. V. D., Roozendael, M. V., Vlemmix, T., Vonk, J., Wagner, T., Wang, Y., Wang, Z., Wenig, M., Wiegner, M., Wittrock, F., Xie, P., Xing, C., Xu, J., Yela, M., Zhang, C., and Zhao, X.: Intercomparison of MAXDOAS vertical profile retrieval algorithms: Studies on field data from the CINDI-2 campaign, Atmospheric Measurement Techniques, 14, https://doi.org/10.5194/AMT-14-1-2021, 2021.

*Line 126. List specific AERONET aerosol retrieval results extrapolated to the matching windows and discuss the extrapolation method.*

*-Which AERONET wavelengths were used in the calculation of Angstrom Exponent?*

*- For this application, AERONET shortest wavelengths (340 and 380 nm) should have been used.*

*- AERONET does not retrieve single scattering albedo at wavelengths shorter than 440 nm. How was aerosol absorption accounted for in the UV region?*

**We updated the text as follows:**

**"In this study, aerosols have been retrieved from the $O_4$ absorption band at 360 nm. Using the mean Angstrom Exponent 440 - 870nm from AERONET Inversion data (Version 3, Almucantar, Level 1.5), the retrieval results have been extrapolated to the matching spectral windows of HCHO and CHOCHO, as the AERONET station only retrieves data at 440nm or higher wavelengths. AERONET Inversion data was also used for the aerosol parametrization within BOREAS. The AERONET 440 nm single scattering albedo (SSA) and phase function obtained at the time closest to the corresponding slant column measurement scan have been used within the profile inversion (440 nm products are the closest available to 360 nm wavelength products)."**

**We expect that using a phase function retrieved at 440nm will perform better than, for instance, a Henyey-Greenstein phase function with an asymmetry factor extrapolated from higher wavelengths to 360nm.**

*Line 160. CALIPSO or icesat-2 lidar data may have been available on this day. If so, how does it compare?*

*Line 173. Can these results be confirmed with CALIPSO or icesat2 lidar observations? CALIPSO data may be affected by the South Atlantic Anomaly. Icesat2, however, reports observations over the analysis period in this paper at https://icesat-2.gsfc.nasa.gov/atmo-data.*

**We would like to thank the Referee for pointing out the source of this data. This analysis was included in an Appendix (Appendix A: ICESat-2 observations of the biomass burning event) showing ATLAS results for November 23 and 24 over Uruguay. The data is in agreement with the presence of the BB plume at the heights discussed in the manuscript.**

*Line 175. Discuss the expected lifetimes of the retrieved species. It may be relevant in this analysis.*

**$NO_2$, CHOCHO, and HCHO have inherently short atmospheric lifetimes due to rapid removal by photolysis and reactions with hydroxyl radicals (OH). These lifetimes typically range from a few hours (~2 to 4 hours). However, continued emissions from fires, particularly of trace gases like CHOCHO and HCHO, can contribute to their apparent lifetime extension within the mixing layer in present of aerosols, resulting in lifetimes reaching up to ~29 hours (Alvarado et al., 2020). We consider that this is a point from which future work is needed and that enhances the importance of this study, especially in South America. However, more data may be needed.**

*Line 182. Resolve QDOAS acronym.*

**QDOAS is the name of the software used for the slant column densities retrieval using DOAS method. We change the phrase where QDOAS is presented as follows (from line 182):**

**"The evaluations were performed using QDOAS software…"**

*Line 182. Discuss Tables 1 and 2 separately. They provide different information. Describe each column of Table 1 in detail. No absorption cross-section data is shown in Table 2.*

**We combined tables 1 and 2 into one (new Table 1) with all relevant information. This table shows the configuration parameters to carry out the DOAS evaluation procedure. Absorption cross section data for each trace gas are indicated by the corresponding reference; the cross sections files were downloaded from the MPI-Mainz UV/VIS Spectral Atlas, cited as Keller-Rudet et al., 2013 in the manuscript.**

*Line 189 The description of Figure 5 is incomplete. Only the top row is discussed. There is not an adequate discussion of the bottom three rows. The figure caption should accurately describe what is shown in each of the nine panels of the figure. If some of the data shown in this figure is not relevant to the analysis, then remove it.*

**We modified Figure 5 legend as:**

"**Figure 5. Example of DOAS fits for a spectrum measured during the event on November 24 at 13:04 (LT), recorded at 5 degree elevation angle pointing towards west. Blue lines are the optical densities associated with each trace gas and the orange lines are the fitted differential absorption cross sections. The retrieved dSCD and their fitting errors are displayed in molec. $cm^{-2}$ for HCHO, CHOCHO, $NO_2$ and $H_2O$, and in $molec^2$. $cm^{-5}$ for $O_4$. On the last row the residual of DOAS evaluation for each analysis spectral window used is shown.**"

**We also added the following text (from line 194) to improve the description of the corresponding Figure:**

"**Figure 5 shows an example of DOAS analysis for each fitting interval, performed on a spectrum acquired on November 24, 2020 at 13:04 local time (LT). The first row shows the fitting of the target gases, HCHO, $O_4$ and CHOCHO. Other trace gases exhibiting absorption in the same spectral interval, as well as the synthetic Ring spectrum, are shown in the second and third rows. The residual of the analysis is shown in the last row.**

**In this example, the DOAS fits reveal a strong formaldehyde signal, in contrast to what usually appears in Montevideo. In addition, glyoxal was also observed, which had never been detected previously in the same location. A strong signal was also observed for $NO_2$. However, this is not necessarily attributed to the influence of the event over Montevideo. Detailed discussion about this point is presented in Section 4.**"

*Line 193. Describe in some detail the mentioned color index classification algorithm. How is the comparison of intensities used to discern clear from cloudy or partly cloudy conditions?*

**The color index classification algorithm used compares the quotient of intensity measured at 370 nm and 440 nm throughout the day. For each elevation angle, this ratio results in an "U"-inverted characteristic shape when the day is clear. On completely cloudy days, the quotient remains almost constant. If there are scattered clouds, the quotient is highly variable over time.**

**In this way, it is possible to characterize the ratio throughout a full day and train a basic model for classification. This algorithm aims to automate a first-order classification when the data volume is very large. In this study, since the number of dates covered is relatively small, the information on which days were cloudy was also cross-checked with weather station information to validate it.**

**We rephrased this definition as (from line 202):**

"**Cloudy days were screened from our spectral data set using an algorithm based on the diurnal cycle of the color index given by the ratio between intensities at 370 nm and 440 nm (Gielen et al., 2014; Wagner et al., 2014). This ratio is strongly affected by cloudy conditions, and its variation provide a way to flag clear days. November 25 and 28 were**

tagged as completely cloudy and removed from the analysis as a result of this first order data screening. We used data from weather stations to manually confirm this result."

*Line 198. Change the x-axis time representation in Figure 6 to the commonly used month-day nomenclature. The figure caption should clearly indicate what is shown on each row from top to bottom, and on each column from left to right.*

We used the ACP's *Submission Guidelines* recommended format for date and time in Figure 6 (dd-mm). We can change the format upon the Editor's request.

The caption of Figure 6 was changed as follows:

"Differential Slant Column Densities (dSCD) for elevation angles 5° (red), 10° (green) and 20° (blue), for the trace gases HCHO, $O_4$, CHOCHO and $NO_2$ (first to fourth row, respectively) for the study period. Results for east and west directions are separated in the left and right panels, respectively. An increase in the dSCD values of HCHO is observed on November 24. The O4 dSCD values observed for that date are lower and coalesce for the different elevation angles, which may indicate a high aerosol load. The glyoxal signal also increases on November 24, although not as clearly as that of formaldehyde. The observed changes in $NO_2$ dSCD values for that day with respect to background are relatively small and may not be attributed to the event."

We also updated the main text explaining more details about Figure 6 (from line 209):

"The results of the DOAS retrievals for the period around the detection of the event are shown in Figure 6 for the trace gases HCHO, $O_4$, CHOCHO and $NO_2$ (first to fourth rows, respectively). The colored circles correspond to the dSCD values for elevation angles of 5 (red), 10 (green) and 20 (blue) degrees in the east and west observation directions. The $O_4$ dSCDs obtained in the period from November 20 to 28 have a behavior typical of days with clear skies and low aerosol load, except for November 24. On this day, $O_4$ results are unusually low, showing approximately the same values at all elevation angles. This can be interpreted as the result of multiple scattering in the atmosphere, which alters light path lengths and indicates a high aerosol load (Wagner et al., 2004; Sinreich et al., 2013)."

*Line 219. Sentinel 5 (S5P) is just the satellite name. The sensor's name should also be included*

We updated the caption of Fig. 7 as follows:

"Panels (a-c) show the time series of vertical column densities (VCD) of glyoxal, formaldehyde and nitrogen dioxide. Percentile values for the corresponding products from TROPOMI-S5P over Montevideo are also shown as horizontal lines. Panel (d) shows the time series for the retrieved AOD by BOREAS from the MAX-DOAS observations and AERONET AOD values (at 360 and 380 nm, respectively)."

*Line 225. Change the x-axis time representation in Figure 7 to the commonly used month-day nomenclature.*

**We used the ACP's *Submission Guidelines* recommended format for date and time in Figure 6 (dd-mm). We can change the format upon the Editor's request.**

*Line 227. AERONET does retrieve AOD at 360 nm. Clarify in the text, and in the Figure 7 caption the wavelength of the AERONET AOD retrieval. The same comment applies to the description of AERONET AOD shown in Figure 8a.*

**Done.**

**The text now reads (from line 240):**

**"Figure 8(a) shows the AOD on November 24, comparing the retrieval with AERONET values (at 360 and 380 nm, respectively),…"**

**As mentioned before, caption of Figure 7 was modified to make this information clearer.**

*Lines 230-232. This statement makes no sense. AERONET measurements are insensitive to formaldehyde and glyoxal. Please rephrase.*

**To improve the readability, the text was rephrased (from line 243):**

**"The vertical column densities obtained from the BOREAS retrieval show a pronounced increase for formaldehyde and glyoxal on November 24 compared to the values of the other days, which is consistent with the passage of the plume over Uruguay found in satellite images (see Figures 2 and 3). The variability in the median values of TROPOMI-S5P CHOCHO, HCHO and $NO_2$ generally aligns with the corresponding MAX-DOAS measurements for most of compared the days. However, we note that there is a considerable variability in CHOCHO values - similar to MAX-DOAS observations - likely arising from the inherent difficulties in its retrieval."**

*Line 243 Resolve CINDI-II acronym.*

**Done.**

*Line 245. CALIPSO lidar data, if available, should confirm the presence (or not) of aerosols above 4 km.*

**Unfortunately, there is no useful data from CALIPSO for the event over Montevideo for November 24.**

*Line 277. The authors seem to refer to the UV Aerosol Index (UVAI). Since it is the name of a specific satellite product it should be capitalized. The authors should briefly explain why the UVAI readings are relevant in the context of this work and provide relevant references. In addition to the qualitative UVAI product, 388 nm aerosol optical depth and single scattering albedo are also available from TROPOMI observations (Torres et al., AMT, 2020).*

**Thank you for pointing that out. We are referring to the TROPOMI Level 2 Ultraviolet Aerosol Index (UVAI), version 2 (Copernicus Sentinel-5P (processed by ESA), 2021). This data product is available from the European Space Agency (https://sentinel.esa.int/documents/247904/2474726/Sentinel-5P-Level-2-Product-User-Manual-Ozone-Total-Column). We use the UVAI data as a reference for the presence of aerosols in the plume, which likely coincides with the location of the trace gases.**

**The updated text, from line 98, now reads**

**"We employed TROPOMI observations of NO₂ and CO in our analysis to detect long distance transport events. We also used UV Aerosol Index (UVAI) data as a reference for the presence of aerosols in the plume, which likely coincides with the location of the trace gases (see for instance Torres et al., 2020, Alvarado et al., 2020)."**

**and we added the reference to Torres et al., 2020:**

Torres, O., Jethva, H., Ahn, C., Jaross, G., and Loyola, D. G.: TROPOMI aerosol products: evaluation and observations of synoptic-scale carbonaceous aerosol plumes during 2018–2020, Atmos. Meas. Tech., 13, 6789–6806, https://doi.org/10.5194/amt-13-6789-2020, 2020

*Line 283 UV Aerosol Index*

**Done**

*Line 291 UV Aerosol index*

**Done**

*Line 312 Include the corresponding AERONET wavelength.*

**We added the 380 nm wavelength information to the corresponding phrase and the phrase was rewritten:**

**"On November 24, AERONET measured very high AOD values in all bands for Montevideo. For example, there were AOD values at 380 nm greater than 1.0."**

---

## Author Response (AR2)

**We thank Referee #1 for giving us the opportunity to make some points clear. Please find below in blue the responses to the comments.**

*I want to thank the authors for addressing my previous questions/comments.*

*I think the authors should be more clear about the main goal of the paper.*

*In case the main goal is to detect the plume, why not take advantage of satellite data?*

*In case the authors want to identify and quantify the plume, then the combination of the AERONET and MAX-DOAS would be more aligned.*

The main goal of our study was not only to detect, but to identify and quantify the plume. For the detection of the plume, satellite data would probably be sufficient to identify the main aspects such as source and transport of some constituents of the emissions. However, in the case discussed in this manuscript, satellite data was unable to fully describe the increase in aerosol loading, and the presence of HCHO and CHOCHO over Montevideo, which was demonstrated by MAX-DOAS and AERONET measurements. The main reasons are the higher detection limit and the limited time coverage of the satellite data.

Although the simultaneous use of satellite and ground-based measurements seems redundant at first glance, we have attempted to show how combining these complementary measurements provide a better overview of the emissions released during low-intensity biomass combustion and their impact at long distances from the sources.

We think that this is already covered by the last sentences of the abstract:

*"This study underscores the potential of ground-based atmospheric monitoring as a tool for detection of such events. Furthermore, it demonstrates greater sensitivity compared to satellite when it comes to detection of relatively small amounts of carbonyls like glyoxal and formaldehyde."*

*Another point is related with the number of cases the authors presented. Is it possible to apply the methodology proposed in the paper to others plume events?*

It would certainly be possible to apply the proposed methodology to observe other biomass burning events. This depends on the magnitude of the event and the transport direction of the emissions. For example, Alvarado et al., 2020 investigated an intense wildfire event where similar species were transported over Canada, mainly using satellite data. There are also other reports that make use of this kind of methodology to describe the anthropogenic effects on the atmospheric composition of the species considered in this study (e.g. Benavent et al., 2019; Ryan et al., 2023).

In South America, uncontrolled fires of different types of vegetation are frequently observed, especially within the Amazonian rainforest. To the best of our knowledge, this is the first report of low-intensity wildfires in South America described using the combination of tools mentioned above. Gathering information to continue this research is one of the long-term goals of our research group, as these techniques could help fill the knowledge gap in the Southern Hemisphere.

We tried to make clear this point in the summary of the manuscript:

*"Continuous measurements of this type of stations at several locations on the South-American continent would provide valuable information on the regional and continental scale effects of wild fires. They also would deliver the data needed to validate atmospheric models to investigate the chemical processes in the plumes."*

**References:**

Benavent, N., Garcia-Nieto, D., Wang, S., and Saiz-Lopez, A.: MAX-DOAS measurements and vertical profiles of glyoxal and formaldehyde in Madrid, Spain, Atmos. Environ, 199, 2019, 357-367, https://doi.org/10.1016/j.atmosenv.2018.11.047.

Ryan, R. G., Marais, E. A., Gershenson-Smith, E., Ramsay, R., Muller, J.-P., Tirpitz, J.-L., and Frieß, U.: Measurement report: MAX-DOAS measurements characterise Central London ozone pollution episodes during 2022 heatwaves, Atmos. Chem. Phys., 23, 7121–7139, https://doi.org/10.5194/acp-23-7121-2023, 2023.